# Climatic traits on daily clearness and cloudiness indices

Estefanía Muñoz[1] and Andrés Ochoa[1]

[1]Universidad Nacional de Colombia, Medellín

**Correspondence:** Estefanía Muñoz (emunozh@unal.edu.co)

**Abstract.** Solar radiation has a crucial role in photosynthesis, evapotranspiration and other biogeochemical processes. The amount of solar radiation reaching the Earth's surface is a function of astronomical geometry and atmospheric optics. While the first is deterministic, the latter has a random behaviour caused by highly variable atmospheric components as water and aerosols. In this study, we use daily radiation data (1978-2014) from 37 FLUXNET sites distributed across the globe to inspect for climatic traits in the shape of the probability density function (PDF) of the clear-day ($c$) and the clearness ($k$) indices. The analysis was made for shortwave radiation (SW) at all sites and for photosynthetically active radiation (PAR) at 28 sites. We identified three types of PDF, unimodal with low dispersion (ULD), unimodal with high dispersion (UHD) and bimodal, with no difference in the PDF type between $c$ and $k$ at each site. Looking for regional patterns in the PDF type we found that latitude, global climate zone and Köppen climate type have a weak and the Holdridge life a stronger relation with $c$ and $k$ PDF types. The existence and relevance of a second mode in the PDF can be explained by the frequency and meteorological mechanisms of rainy days. These results are a frame to develop solar radiation stochastic models for biogeochemical and ecohydrological modeling.

## 1 Introduction

Solar radiation drives most physical, chemical and biological processes at the earth's surface. It is the primary energy source for photosynthesis, evapotranspiration and other biochemical processes (Wu et al., 2016; Mercado et al., 2009). The amount of solar irradiance reaching any place on the Earth's surface at a given time results from Sun's emission spectrum, Sun-Earth distance, the angle of incidence of solar rays, and the atmospheric attenuation of light. The geometry of Earth's orbit and rotation is well-known and can be calculated with high precision. However, atmospheric attenuation of light is strongly affected by atmospheric constituents such as molecular gases, aerosols, water vapor and clouds by reflecting, absorbing, and scattering processes (Platt et al., 2012; Wallace and Hobbs, 2006). Aerosols, water vapor and clouds are highly variable in space and time. As a consequence, uncertainty is unavoidable when calculating surface solar radiation due to the high space and time variability of aerosols, water vapor and clouds (Li and Trishchenko, 2001; Chen et al., 2000).

Scientists have affronted the problem of atmospheric light attenuation by mechanistic and statistical approaches. While the former deal with the physical and chemical processes governing light attenuation, the latter use large amounts of observations to infer patterns of variability caused. Two indices are widely used to quantify the random nature of atmospheric light attenuation, the clear-sky index ($c$) (see Tran, 2013; Harrouni, 2008; Ianetz and Kudish, 2008; Allen et al., 2006; Hansen, 1999; Skartveit

and Olseth, 1992; Bendt et al., 1981; Gordon and Hochman, 1984; Liu and Jordan, 1960), defined as the ratio of actual radiation to clean-dry atmosphere radiation, and the clearness index ($k$) (see Engerer and Mills, 2014; Hollands and Suehrcke, 2013; Ianetz and Kudish, 2008; Polo et al., 2008; Olseth and Skartveit, 1984) which is the ratio of actual radiation to top-of-the-atmosphere radiation (i.e. with no atmospheric attenuation). Although $c$ and $k$ can be calculated for any spectral band and time aggregation scale, they are often studied at the hourly or daily time steps and for the shortwave band (e.g. Utrillas et al., 2018; Cañada et al., 2003; Martinez-Lozano et al., 1999). Notice that because of the different physical mechanisms involved in the magnitude, frequency and duration of clouds, water vapor and aerosols across the globe, $c$ and $k$ must show statistical properties related to regional/local climate.

In this paper, we analyze the statistical properties of $c$ and $k$ at 37 FLUXNET sites distributed worldwide (section 2) looking for climate-related variability patterns. We process historical extraterrestrial spectral irradiance data from the SOLID project (section 2) by using mechanistic models –solar geometry and the Beer–Lambert law (section 3)– to remove the deterministic component of historical daily solar radiation observations. Then, we analyze the probability distribution function of $c$ and $k$ in relation to global climate regions, Köppen climate types and Holdridge life zones (sections 4 and 5).

Characterizing the stochastic behaviour of surface solar radiation is of great importance in many research fields, e.g. photovoltaic electricity generation, photosynthesis, nutrient dynamics in ecosystems, water dynamics in soils and forest fire risks (e.g. Muñoz et al., 2018; Engerer and Mills, 2014). Of special interest are the ecohydrological and biochemical models of (Rodríguez-Iturbe and Porporato, 2004) and collaborators (e.g. Tamea et al., 2011; Laio et al., 2009; Nordbotten et al., 2007; Manzoni et al., 2004; Porporato et al., 2003), which have had a great progress since the beginnings of the 20th Century. By solving the water balance equation for the rooting soil depth at a daily time-step, they analyze the dynamics of soil moisture (Rodríguez-Iturbe et al., 1999b; D'Odorico et al., 2000; Porporato et al., 2002). In their approach, the water balance equation is a stochastic differential equation forced by the daily rainfall stochastic process. As a result, rainfall probabilistic structure propagates to all other variables of the system. Rodríguez-Iturbe and his coauthors have used this model to study the dynamics of water table depth (Ridolfi et al., 2008; Laio et al., 2009; Tamea et al., 2009), runoff (Botter et al., 2008; Ceola et al., 2010), leaching (Botter et al., 2007), vegetation competition and colonization (Rodríguez-Iturbe et al., 1999a; Fernández-Illescas and Rodríguez-Iturbe, 2003), soil carbon and nitrogen (D'Odorico et al., 2003; Porporato et al., 2003; Ridolfi et al., 2003; Manzoni et al., 2004), vegetation water stress (Ridolfi et al., 2000; Rodríguez-Iturbe et al., 2001; Laio et al., 2001; Porporato et al., 2001), photosynthesis (Daly et al., 2004a, b), plant biomass (Schaffer et al., 2015; Nordbotten et al., 2007), waterborne human pathogen invasions (Gatto et al., 2013) and soil related phytopathology (Thompson et al., 2013) in water-controlled ecosystems. All these remarkable works, however, have been oriented to water-limited ecosystems, where uncertainty is introduced by the rainfall process only. To study energy-limited ecosystems, daily solar radiation is required as one more external variable driving evaporation and transpiration processes. Solar radiation is also a random variable, since it highly depends on atmospheric transmittance, specially that of clouds (Muñoz et al., 2020). This paper has the purpose of establishing a framework for daily solar radiation characterization that serves as base for developing the ecohydrology of energy-limited ecosystems.

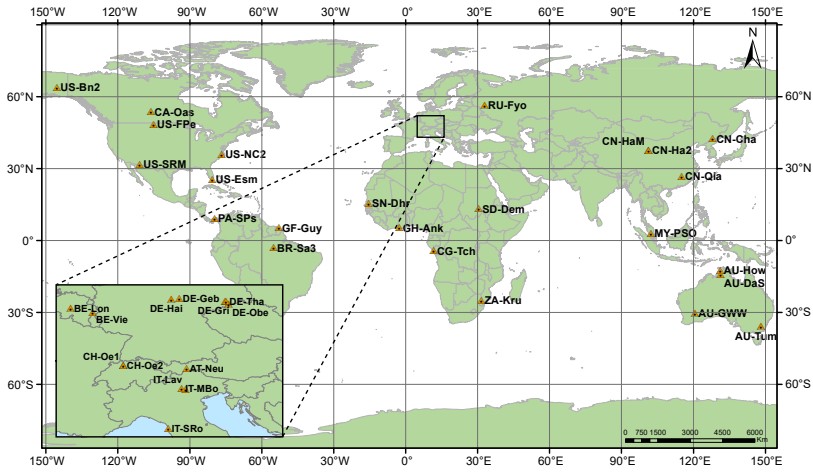

**Figure 1.** The sites selected from the FLUXNET dataset are spanned over several continents and climates.

## 2 Data

Our data set consists of daily observations of incoming solar radiation and rainfall from 37 sites around the world from the FLUXNET data set (Baldocchi et al., 2001; Olson et al., 2004) (Fig. 1). We analyze two spectral bands, the photosynthetically active radiation (PAR) and the shortwave radiation(SW). While SW observations are available at all sites, PAR observations are available at 28 sites only. Sites have different periods of record spanning from 1996 to 2014 and elevations from sea level to 1550 m (Table 1). These sites were selected from an initial set of more than 200 sites after filtering by several criteria as record length, data quality and spatial coverage of the whole group. FLUXNET PAR data are given as photosynthetic photon flux density (PPFD). The wavelength domain for PPFD in the FLUXNET data set is 400–700 nm (Olson et al., 2004) and has units of $\mu$ mol m$^{-2}$ s$^{-1}$. We convert PPFD to PAR irradiance in W m$^{-2}$ through the relationship 4570 nmol m$^{-2}$ s$^{-1}$=1 W m$^{-2}$ (Sager and McFarlane, 1997).

We use the Solar Spectral Irradiance (SSI) at the top of the atmosphere from the "First European Comprehensive Solar Irradiance Data Exploitation project" (SOLID) (Haberreiter et al., 2017; Schöll et al., 2016) as input data for an atmospheric radiation transfer model (section 3). The SOLID spectral time series has a daily time resolution from 1978/7/11 to 2014/12/31 (13204 days) and covers the wavelength range between 0.5 and 1991.5 nm. Data from SOLID are available at http://projects. pmodwrc.ch/solid.

In order to analyze the spatial climatic patterns of the random component of PAR and SW radiation, we use the Köppen climate classification from Peel et al. (2007), downloaded from the author's webpage and the Holdridge life zones (Holdridge, 1947, 1967) from Leemans (1992), downloaded from UNEP-WCMC.

**Table 1.** FLUXNET Sites. The record period refers to complete calendar years, i.e. data for all sites start on January 1st of the initial year and end on December 31st of the last year.

| Site | Country | Latitude [°] | Longitude [°] | Elevation [m] | Period |
|------|---------|-------------|--------------|--------------|--------|
| AT-Neu | Austria | 47.117 | 11.318 | 970 | 2002–2012 |
| AU-DaS | Australia | -14.159 | 131.388 | 53 | 2008–2014 |
| AU-GWW | Australia | -30.191 | 120.654 | 486 | 2013–2014 |
| AU-How | Australia | -12.494 | 131.152 | 41 | 2001–2014 |
| AU-Tum | Australia | -35.657 | 148.152 | 1249 | 2001–2014 |
| BE-Lon | Belgium | 50.552 | 4.746 | 167 | 2004–2014 |
| BE-Vie | Belgium | 50.305 | 5.998 | 493 | 1996–2014 |
| BR-Sa3 | Brazil | -3.018 | -54.971 | 100 | 2000–2004 |
| CA-Oas | Canada | 53.629 | -106.198 | 530 | 1996–2010 |
| CG-Tch | Congo | -4.289 | 11.656 | 82 | 2006–2009 |
| CH-Oe1 | Switzerland | 47.286 | 7.732 | 450 | 2002–2008 |
| CH-Oe2 | Switzerland | 47.286 | 7.734 | 452 | 2004–2014 |
| CN-Cha | China | 42.402 | 128.096 | 761 | 2003–2005 |
| CN-Ha2 | China | 37.609 | 101.327 | 3190 | 2003–2005 |
| CN-HaM | China | 37.370 | 101.180 | 3250 | 2002–2004 |
| CN-Qia | China | 26.741 | 115.058 | 79 | 2003–2005 |
| DE-Geb | Germany | 51.100 | 10.9143 | 161.5 | 2001–2014 |
| DE-Gri | Germany | 50.950 | 13.513 | 385 | 2004–2014 |
| DE-Hai | Germany | 51.079 | 10.453 | 430 | 2000–2012 |
| DE-Obe | Germany | 50.787 | 13.721 | 734 | 2009–2014 |
| DE-Tha | Germany | 50.962 | 13.565 | 385 | 1996–2014 |
| GF-Guy | French Guiana | 5.279 | -52.925 | 48 | 2004–2014 |
| GH-Ank | Ghana | 5.268 | -2.694 | 124 | 2011–2014 |
| IT-Lav | Italy | 45.956 | 11.281 | 1353 | 2003–2014 |
| IT-MBo | Italy | 46.015 | 11.046 | 1550 | 2003–2013 |
| IT-SRo | Italy | 43.728 | 10.284 | 6 | 1999–2012 |
| MY-PSO | Malaysia | 2.973 | 102.306 | 147 | 2003–2009 |
| PA-SPs | Panama | 9.314 | -79.631 | 68 | 2007–2009 |
| RU-Fyo | Russia | 56.461 | 32.922 | 265 | 1998–2014 |
| SD-Dem | Sudan | 13.283 | 30.478 | 500 | 2005–2009 |
| SN-Dhr | Senegal | 15.403 | -15.432 | 40 | 2010–2013 |
| US-Bn2 | USA | 63.920 | -145.378 | 410 | 2002–2004 |
| US-Esm | USA | 25.438 | -80.595 | 1.07 | 2008–2014 |
| US-FPe | USA | 48.308 | -105.102 | 634 | 2000–2008 |
| US-NC2 | USA | 35.803 | -76.668 | 5 | 2005–2010 |
| US-SRM | USA | 31.821 | -110.866 | 1120 | 2004–2014 |
| ZA-Kru | South Africa | -25.020 | 31.497 | 359 | 2009–2013 |

## 3  Methods

Daily radiation amount at a site on the earth's surface is the result of integrating instantaneous irradiance over the day length. Surface instantaneous irradiance estimation comprises solar irradiance at the top of the atmosphere (TOA) and the physical properties of the atmosphere for the site and time of interest. We use SOLID data for TOA irradiance and the Beer–Lambert law to calculate light attenuation by the atmosphere (see details in section sec:Hcda). However, some atmospheric components as clouds, water vapor, and aerosols are highly variable in space and time, which is troublesome when using the Beer–Lambert law for a one-layer atmosphere. Therefore, we follow two approaches: 1) use the clear-day index ($c$) (also known as relative clearness index, clear day index, and normalized clearness index) to assess the effect of the variable components on total daily radiation, and 2) use the clearness index ($k$) to assess the whole atmospheric effect on total daily radiation. Both indices are defined in Eqs. (1), where $H_{obs}$ is the observed daily global radiation on a horizontal surface at the ground level, $H_0$ is the extraterrestrial daily global radiation on a horizontal surface, and $H_{cda}$ is the daily global radiation on a horizontal surface on

the ground for a cloudless, clean, and dry atmosphere.

$$c = \frac{H_{obs}}{H_{cda}}, \quad k = \frac{H_{obs}}{H_0} \tag{1}$$

## 3.1 Daily radiation at the top of the atmosphere

Integration of the daily $SSI$ over the spectral band of interest (400–700 nm for PAR, 285–280 nm for SW) gives the Total Spectral Irradiance for each band ($TSI_{PAR}$ and $TSI_{SW}$) at TOA, as shown in Eq. (2). After some geometric transformations accounting for solar declination ($\delta$) and latitude ($\phi$) (Iqbal, 1983), the daily globla radiation on a horizontal surface can be calculate as shown in Eq. (3).

$$TSI_{Band} = \int_{Band} SSI \, d\lambda \tag{2}$$

$$H0_0 = \frac{24 E_0}{\pi} TSI_{Band} \left( \omega_{sr} \sin\delta \sin\phi + \cos\delta \cos\phi \sin\omega_{sr} \right) \tag{3}$$

where $E_0$ is the eccentricity correction factor of the earth's orbit and $\omega_{sr}$ is the sunrise hour angle for the day.

## 3.2 Daily surface radiation for a cloudless, clean dry atmosphere

Daily surface radiation for a cloudless, clean, and dry atmosphere ($H_{cda}$) is the sum of the direct ($H_b$) and diffuse ($H_d$) components. To calculate daily $H_b$ and $H_d$ on a horizontal surface at the ground level, we model the direct and diffuse instantaneous spectral irradiances and integrate them along the day length and the PAR and SW spectral domains.

Following Iqbal (1983), we assume the cloudless, clean and, dry atmosphere to be composed by uniformly mixed gases (m) and ozone (o). Using the Beer–Lambert law and integrating, daily $H_b$ is calculated as in Eq. (4).

$$R_b = \int_{\gamma_{sr}}^{\gamma_{ss}} \int_{Band} SSI_{0,n,\lambda} E_0 \sin(\gamma) \tau_{ma,\lambda} \, d\lambda \, d\gamma \tag{4}$$

where $SSI_{0,n,\lambda}$ is the extraterrestrial spectral irradiance normal to the rays from the sun (obtained from SOLID), $\gamma$ is the solar altitude varying from sunrise ($sr$) to sunset ($ss$), and $\tau_{ma,\lambda}$ is the transmittance due to the molecular absorbers of the cda atmosphere.

For the assumed atmosphere composition $\tau_{ma,\lambda} = \tau_o \cdot \tau_g$, where $\tau_o$ and $\tau_g$ are the ozone and the mixed gases transmittance, respectively (see details in Iqbal, 1983, Sec.6.14). We arbitrarily assumed forward and backward scatterances of $0.5$ and considered only the first pass of radiation through the atmosphere. Although higher reflectances could bring about some subes-

timation of $H$, especially during snow-cover periods, we think it is not a critical issue for the sake of this study. Uncertainty caused by these two assumptions will be included in the statistical properties of $c$. $H_d$ can then be calculated by

$$H_d = \int\limits_{\gamma_{sr}}^{\gamma_{ss}} \int\limits_{Band} SSI_{0,n,\lambda} E_0 \sin(\gamma)\tau_{ma,\lambda}[0.5(1-\tau_{r,\lambda})] \, \mathrm{d}\lambda \, \mathrm{d}\gamma \tag{5}$$

where $\tau_{r,\lambda}$ is the transmittance due to Rayleigh molecular scattering (see details in Iqbal, 1983, Sec.6.14).

5    Several atmospheric parameters are required by Eqs. (4) and (5). We assume the 1976 U.S. standard atmosphere (NASA, 1976) (sea level pressure of 101.325 kPa, sea level temperature of 288 K, and sea level density of 1.225 kg/m$^3$) and the Kasten and Young (1989, Table II) optical air mass function of solar altitude, which has 336 values for solar altitudes between 0°and 90°. Transmittance for ozone and mixed gases are calculated as in Eqs. (6) to (8).

$$\tau_{o,\lambda} = \exp(-k_{o,\lambda} l_o m_r) \tag{6}$$

$$\tau_{g,\lambda} = \exp\left[\frac{-1.41 k_{g\lambda} m_a}{(1+118.93 k_{g\lambda} m_a)^{0.45}}\right] \tag{7}$$

$$\tau_{r,\lambda} = \exp(0.008735\lambda^{-4.08} m_a) \tag{8}$$

where $m_r$ is relative air mass at standard pressure, $m_a$ is relative air mass at actual pressure, $k_o$ and $k_g$ are the absorption atten-
uation coefficient for oxygen and mixed gases, and $l_o$ is the amount of ozone in cm (at normal temperature and pressure, NTP). We calculate $k_{o,\lambda}$ for any $\lambda$ value using the Leckner (1978) interpolation of the classic Vigroux (1953) data. For calculating $l_o$ we interpolate, for each latitude and doy of interest, the Table 5.3.2 from Iqbal (1983) which gives the monthly total amount of ozone in a vertical column of air for several latitudes. Table 5.3.2 from Iqbal (1983) is a reproduction of Robinson (1966, p.114). $k_{o,\lambda}$ is calculated by interpolating Table 6.13.1, which is a reproduction of Table 4 in Leckner (1978, p.146).

## 3.3   Statistical properties of $k$ and $c$

After the process described is sections 3.1 and 3.2, we calculated daily time series of $c$ and $k$ by using the expressions in Eq. (1) for the $SW$ and $PAR$ spectral bands. Then, we estimated the mean annual cycle, and empirical probability density functions (PDF) of $H$, $c$, and $k$ for both bands. We separate the data samples of $c$ and $k$ by humid and dry days, using precipitation data as a proxy of cloudiness and water vapor in the atmosphere. Then, we inspect the seasonality of $c$ and $k$ by comparing
the cumulative distribution function (CDF) of each month with the CDFs of the other months. The comparison of the CDFs is carried out visually and tested by using Kolmogorov–Smirnov (KS) and Anderson–Darling (AD) Goodness of Fit Tests (Dodge, 2008; Pearson, 1900; Scholz and Stephens, 1987). We use both tests because AD is more sensitive to the tails KS to the center of the distribution.

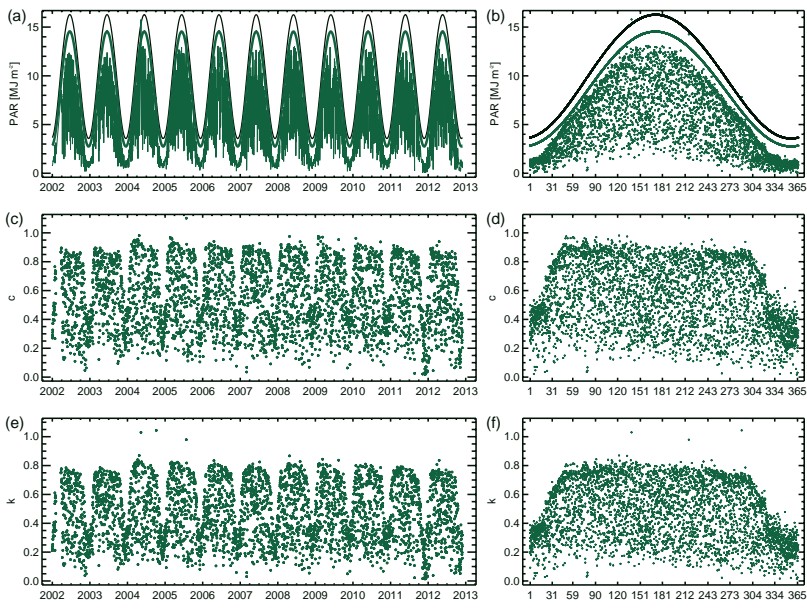

**Figure 2.** Time series and annual cycle of PAR (a–b), $c$ (c–d), and $k$ (e–f) at AT-Neu. Solid black and green lines in panels (a) and (b) indicate $PAR_0$ (no atmosphere), and $PAR_{cda}$ (clean and dry atmosphere), respectively. Thin lines in (a) and points in (b) show $PAR_{obs}$.

## 4   Results

The time series and the mean annual cycle of PAR and SW, $c$ and $k$ were plotted for all sites. Since these are so many figures, they are put in the Supplementary Material. The case of AT-Neu, a site with long and high-quality records, is shown in Fig. 2. A seasonal pattern is observed in the maximum values along the year, but $c$ and $k$ take values spanning over their whole domain.

5    Splitting the data in two samples, the rainy days and dry days, allowed the estimation of the PDFs and CDFs. As shown in Fig. 3, AT-Neu PDFs have a bimodal shape (i.e. a PDF with two modes). Inspection of the PDFs of all sites motivated us to define three types of PDF according to the shape of the function. We propose the following PDF type: Unimodal with low dispersion (ULD), unimodal with high dispersion (UHD) and bimodal (B) (see a graphical description in Fig. 5a).

After separating data by rainy and dry days, we divided them by months to define the seasons of the $c$ and $k$, i.e., groups of

10    months where the indices have the same PDF. These definition was performed observing the aggregation of the monthly CDFs and comparing the CDF of each month with the those of the other months with KS and AD Goodness of Fit Tests. Fig. 4 shows the CDF of each month separated by rainy and dry days, and the comparison matrices of the PDF of each month at the AT-Neu site. Figs. S57 to S84 show the results of all sites studied. The separation of seasons is very clear in the extratropical sites, while in tropical sites monthly CDFs are not clearly grouped, and matrices of comparison do not indicate similar behavior among

15    months. AD method results in more clearly defined seasons and with more months that the KS method, suggesting that the most important differences occur in the center of distributions (where KS is more sensitive).

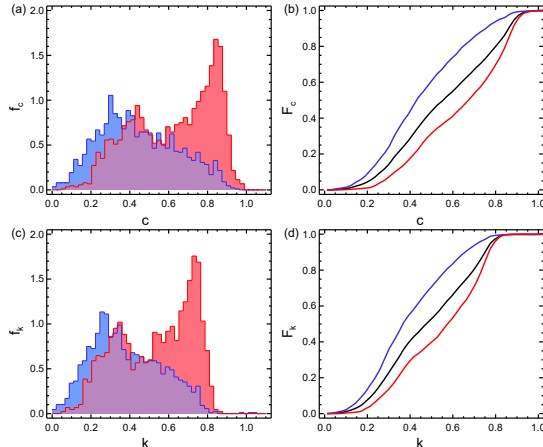

**Figure 3.** PDFs (left panel) and CDFs (right panel) for rainy (blue) and dry (red) days of $c$ (a–b) and $k$ (c–d) for PAR at AT-Neu FLUXNET site.

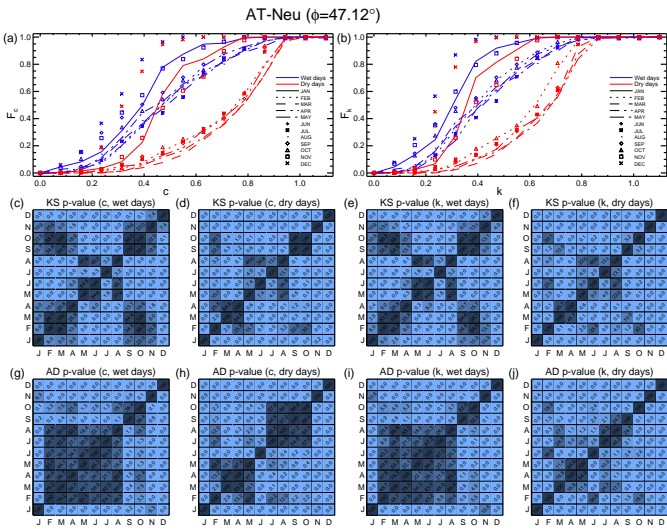

**Figure 4.** Monthly CDFs of $c$ (a), and $k$ (b). p-value of the 2-sample KS and AD tests applied to all combinations of monthly CDF of $c$ and $k$ during wet (blue) and dry (red) days at AT-Neu (c–j). p-values are multiplied by 100 to show more decimals using less space.

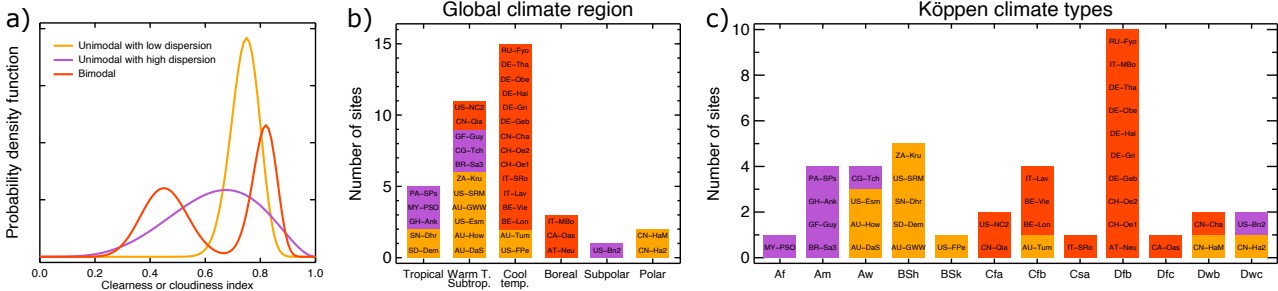

**Figure 5.** Panel a) shows a scheme of the three PDF types of $k$ and $c$. Panels b) and c) show the climatic distribution of the identified PDF types for the 37 case studies.

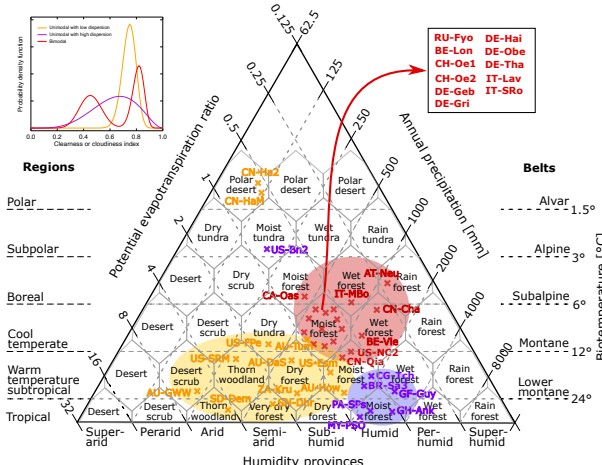

**Figure 6.** Plotting all sites in the Holdridge's life zones triangle shows three groups of sites with the same $c$ and $k$ PDF shape. Red symbols represent bimodal, purple symbols unimodal with high dispersion and yellow symbols unimodal with low dispersion PDFs.

Looking for climate-related regionalization patterns, the $c$ and $k$ PDF type was compared to the global climate region (Fig. 5b), the Köppen climate classification (Fig. 5c) and the Holdridge life zones (Fig. 6). An in-depth inspection of the plots of all sites allowed to set out the following statements: a) the same behavior is observed for $c$ and $k$ at each site, b) latitude is not enough to explain the shape of the PDFs, c) Köppen climate types show a more clear pattern than global climate regions, d) Holdridge life zones show the clear cut pattern of variability of the PDF types. Table 2 summarizes these results and the Köppen and Holdridge classification of each site.

## 5  Discussion

We analyzed the stochastic behavior of the daily clearness and clear-sky indices for the PAR and SW spectral domains. Both indices remove the astronomical seasonality, and $c$ also removes the seasonality of the clean and dry air optical mass. Therefore,

**Table 2.** Climate classification and $k$ and $c$ PDF type of all sites. Table is sorted by decreasing latitude.

| Site | Latitude [°] | Global Climate Region | Köppen climate type | Holdridge Life Zone | PDF Type |
|------|-------------|----------------------|--------------------|--------------------|----------|
| US-Bn2 | 63.920 | Subpolar | Dwc | Alpine Moist tundra | UHD |
| RU-Fyo | 56.461 | Cool temperate | Dfb | Montane Moist forest | B |
| CA-Oas | 53.629 | Boreal | Dfc | Subalpine Moist forest | B |
| DE-Geb | 51.100 | Cool temperate | Dfb | Montane Moist forest | B |
| DE-Hai | 51.079 | Cool temperate | Dfb | Montane Moist forest | B |
| DE-Tha | 50.962 | Cool temperate | Dfb | Montane Moist forest | B |
| DE-Gri | 50.950 | Cool temperate | Dfb | Montane Moist forest | B |
| DE-Obe | 50.787 | Cool temperate | Dfb | Montane Moist forest | B |
| BE-Lon | 50.552 | Cool temperate | Cfb | Montane Moist forest | B |
| BE-Vie | 50.305 | Cool temperate | Cfb | Montane Wet forest | B |
| US-FPe | 48.308 | Cool temperate | BSk | Montane Dry tundra | ULD |
| CH-Oe1 | 47.286 | Cool temperate | Dfb | Montane Moist forest | B |
| CH-Oe2 | 47.286 | Cool temperate | Dfb | Montane Moist forest | B |
| AT-Neu | 47.117 | Boreal | Dfb | Subalpine Rain forest | B |
| IT-MBo | 46.015 | Boreal | Dfb | Subalpine Wet forest | B |
| IT-Lav | 45.956 | Cool temperate | Cfb | Montane Moist forest | B |
| IT-SRo | 43.728 | Cool temperate | Csa | Montane Moist forest | B |
| CN-Cha | 42.403 | Cool temperate | Dwb | Montane Wet forest | B |
| CN-Ha2 | 37.609 | Polar | Dwc | Alvar Polar desert | ULD |
| CN-HaM | 37.370 | Polar | Dwb | Alvar Polar desert | ULD |
| US-NC2 | 35.803 | Warm temperature/Subtropical | Cfa | Lower montane Moist forest | B |
| US-SRM | 31.821 | Warm temperature/Subtropical | BSh | Lower montane Thorn woodland | ULD |
| CN-Qia | 26.741 | Warm temperature/Subtropical | Cfa | Lower montane Moist forest | B |
| US-Esm | 25.438 | Warm temperature/Subtropical | Aw | Lower montane Moist forest | ULD |
| SN-Dhr | 15.403 | Tropical | BSh | - Very dry forest | ULD |
| SD-Dem | 13.283 | Tropical | BSh | - Thorn woodland | ULD |
| PA-SPs | 9.314 | Tropical | Am | - Moist forest | UHD |
| GF-Guy | 5.279 | Warm temperature/Subtropical | Am | Lower montane Wet forest | UHD |
| GH-Ank | 5.268 | Tropical | Am | - Moist forest | UHD |
| MY-PSO | 2.973 | Tropical | Af | - Moist forest | UHD |
| BR-Sa3 | -3.018 | Warm temperature/Subtropical | Am | Lower montane Moist forest | UHD |
| CG-Tch | -4.289 | Warm temperature/Subtropical | Aw | Lower montane Moist forest | UHD |
| AU-How | -12.494 | Warm temperature/Subtropical | Aw | Lower montane Dry forest | ULD |
| AU-DaS | -14.159 | Warm temperature/Subtropical | Aw | Lower montane Dry forest | ULD |
| ZA-Kru | -25.020 | Warm temperature/Subtropical | BSh | Lower montane Dry forest | ULD |
| AU-GWW | -30.191 | Warm temperature/Subtropical | BSh | Lower montane Desert scrub | ULD |
| AU-Tum | -35.657 | Cool temperate | Cfb | Montane Moist forest | ULD |

$c$ is neater than $k$ in describing the effect of the highly variable components of the atmosphere, i.e., clouds, water vapor, and aerosols. Due to the multiple reflections of light by snow, we found a few values of $c$ and $k$ greater than 1 in sites where there is seasonal snow, during the periods in which it occurs (e.g. AT-Neu, Fig. 4). This could introduce important errors if we were performing a forecast work, but it is not so problematic in this study. The analysis of rainy and dry days revealed that $c$ and $k$ have the same PDF shape for both indices at each site. This result is obvious because the whole data sample was separated by rainy and dry days and both indices are designed to take into account the effect of water in the atmosphere. This is also a reason explaining that the PDF shows the same shape type for the SW and the PAR bands.

Although radiation itself is a proxy of cloudiness (Nyamsi et al., 2019; Oliphant et al., 2006), we use rainfall to separate the data sample into rainy and dry days keeping in mind the connection of our stochastic description of attenuation with the family of stochastic models of (Rodríguez-Iturbe and Porporato, 2004) and others. This way, bimodal distributions of $c$ and $k$ vanish in most of the cases when data are divided into rainy and dry days, reinforcing the idea that the modes are strongly related to

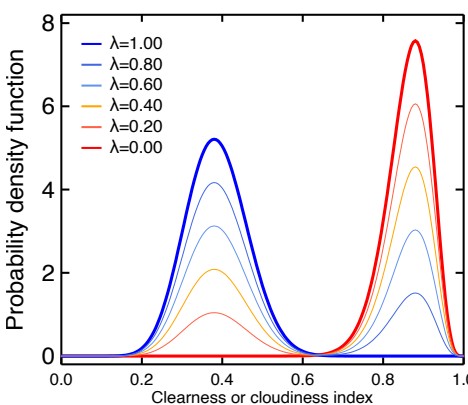

**Figure 7.** A family of PDFs from the general model of Eq. (9).

clear and overcast skies conditions. Exceptions to this pattern occur in AT-Neu (Fig. 4), DE-Geb, and DE-Hai, where the PDFs of dry days still have a bimodal distribution.

Results suggest the occurrence of the three PDF types described in section 4. Our interpretation is that a general bimodal-shaped PDF could explain the three types. Following the nomenclature used in the ecohydrological models of (Rodríguez-Iturbe and Porporato, 2004) and collaborators (e.g. Tamea et al., 2011; Laio et al., 2009; Nordbotten et al., 2007; Manzoni et al., 2004; Porporato et al., 2003), let $\lambda$ be the probability of occurrence a rainy day. Let also $x$ represent any of our indices $c$ and $k$. The PDF of $x$ can then be written as in Eq. (9).

$$f(x) = \lambda f_R(x) + (1 - \lambda) f_D(x) \tag{9}$$

where $f_R(x)$ and $f_D(x)$ are the conditional PDF of $x$ for rainy and dry days respectively. The family of this PDFs shown in Figure 7 has members with one and two modes. The UHD-type PDF emerges in the marginal function $f(x)$ when the two modes of $f_R(x)$ and $f_D(x)$ are close to each other.

The general model of Eq. (9) is also useful to understand the climate influence on the marginal PDF of $c$ and $k$. If $\lambda \to 0$ (i.e. in very dry sites) the $c$ and $k$ PDFs are of the ULD-type, and are the result of solar geometry and the permanent constituents of the atmosphere. If $\lambda$ is not so low, the PDF will be of the UHD-type where convection is the main driver of cloud formation and of the B-type where cloud dynamics is controlled by larger-scale phenomena as atmospheric jets and meteorological fronts (Boucher et al., 2013). Obviously, as $\lambda \to 1$ the $c$ and $k$ PDFs will approximate those of the humid days, no matter the physical mechanisms behind.

## 5.1 Regionalization

As shown in Table 2 and Figure 5, latitude and global climate regions have a weak relation with the PDF types., with ULD occurring in the Tropical, Warm Tropical, Subtropical and Cool temperate regions, UHD in tropical ans subtropical regions,

and B-type PDFS in Warm Tropical, Subtropical, Cool temperate regions and boreal regions. There is also one UHD site in the Subpolar region and two ULD in the Polar region. In summary, there is no clear pattern of PDF type variability when grouped by latitude or global climate regions.

The Köppen classification system performs better than global climate regions to capture the PDF type variability (Figure **??**). C climates (Cfa, Cfb, Csa) and Df climates (Dfb and Dfc) have a clear predominance of the B-type PDFs, with the exception of one site in Australia (AU-Tum). B climates (BSh and BSk) and Aw climates (with one exception) are all of the ULD type. Am climate is UHD, and a variety of PDF types occurs in the Dwb and Dwc climates.

Plotting the sites on the Holdridge life zones scheme (see Fig. 6) reveals a clear-cut pattern of $c$ and $k$ PDF types. The Holdridge life zones triangle has the advantage of showing several climate variables in an independent way. These variables are humidity, annual precipitation and potential evapotranspiration ratio. Boreal and cool temperate regions in the more humid provinces have PDFs of the B type, tropical regions in the humid provinces show PDFs of the UHD type, and the dry provinces show all ULD-type PDFS. Two isolated sites occur at the top of the triangle, two ULD sites in the polar desert of China and one UHD site in Alaska (see map in Figure 1).

## 6  Conclusions

We inspected 37 sites worldwide for the influence of local climate on statistical properties of clearness and clear-day indices. We identified three types statistical behavior according to the PDF shape, namely ULD, UHD and B. The same PDF type was found to occur for $c$ and $k$ at each site for both PAR and WS radiation bands. It was evidenced that latitude is not enough to explain the shape of the PDFs, suggesting that climate could play an important role. Global climate region and the Köppen climate have a stronger relation than latitude with the PDF types. Holdridge life zones classification showed the most clear-cut pattern of variability of the PDF types. We proposed a general mathematical model for the PDF of $c$ or $k$ that has the ULD, UHD and B types as particular cases. These model constitutes an important basis for biogeochemical modeling in energy-limited ecosystems, especially in the field of ecohydrology, but also in meteorology, glaciology, agroclimatology and other research fields.

*Author contributions.*  EM and AO conceived the idea. AO calculated the clear-sky daily radiation. EM did all the statistics. Both EM and AO analyzed the results and wrote the manuscript. AO supervised the work.

*Acknowledgements.*  We thank Departamento Administrativo de Ciencia, Tecnología e Investigación de Colombia (Colciencias) and Universidad Nacional de Colombia for financial support through the programs "Becas de Doctorado Nacionales" and "Convocatoria para el Apoyo al Desarrollo de Tesis de Posgrado de la Universidad Nacional de Colombia 2018", respectively. We also acknowledge the FLUXNET community and the SOLID Project for sharing the data we used in this study.

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
