# Peer review of "Climatic traits on daily clearness and cloudiness indices"

_Biogeosciences, 2020_

## Short Comment (SC1) · 24 Apr 2020

*Conclusions* section heading is missing. It should be before line 9 of page 12.

---

## Referee Comment (RC1) · Anonymous Referee #2 · 4 Jun 2020

Munoz and Ochoa explore patterns of PAR across different latitudes and climate zones. The analysis may be important to the extent that it helps organize and communicate variability in photosynthetically active radiation to the biogeosciences community. At the moment it does not, but I feel that it might. Namely, if the amount of variation in PAR, c, and k explained by solar geometry (obvious) and climate (less obvious) could be determined I could see how obvious aspects of the manuscript could be placed in the context of information that could be quite enlightening for our understanding of how light reaches the surface across the globe. If the authors can do this I feel that the manuscript could be acceptable for publication; at the moment the findings are largely either obvious or unclear, but the latter can be fixed by restructuring the

manuscript and explaining more clearly what was done and its implications. Regarding "Attenuation of light throughout the atmosphere can be calculated by using an attenuation law (e.g. the Beer–Lambert law), but this requires to know the atmospheric optical depth, which is seldom the case" it is also important to note comprehensive atmospheric modeling that seeks to understand the dynamics of atmospheric transmissivity, reflectivity, and absorptivity as a function of wavelength and layer of the atmosphere. Such models are great but difficult to implement at large scales. Page 1 Line 22: light attenuation is not random, it is a function of the physics of the atmosphere. Page 1 Line 25: More evidence is needed that this is the case in the form of references. The Introduction as a whole was a bit too brief. Specifically the notion that c and k are stochastic needs to be addressed in more detail. In many regions, clouds are rather predictable like in areas where sea breezes create weather systems that are easy to anticipate. Fog is another atmospheric phenomena that is expected and predictable in certain times and certain reasons. I had coastal California ecosystems in mind when writing that but then noted that this paper was published just today. (https://agupubs.onlinelibrary.wiley.com/doi/abs/10.1029/2020GL088428) Please expand the introduction to discuss the variables that change c and k. Note also that PAR and the shortwave bands overlap, but incompletely. If you are studying PAR, simply explain why and what the important differences are. Adding the Holdridge / Koppen zones to Table 1 would be an improvement. Why only these 28 sites? There are a number of high-latitude sites with long-term consistent PPFD, for example. Page 4 Line 6: this is true but requires elaboration: 'troublesome when using the Beer–Lambert law'. It is certainly troublesome if the atmosphere is considered to be one layer because atmospheric attenuation will vary dramatically by layer over time, but a layer-by-layer implementation of the Beer-Lambert Law over short time scales may be quite accurate...but difficult to implement After equation 4: 'transmittance due to molecular absorbers of': please note that this is for the clean and dry atmosphere for this particular calculation ('cda') so that people realize why aerosols and other non-molecular absorbers (and reflecters) are excluded. Why is forward / back scattering of

0.5 assumed? Please elaborate in the text. In equation 5, how much do higher-order reflectances typically contribute? It might not be minor, I'm not sure. In equation 6, how is ozone derived? Is it weighted for its distribution throughout the atmospheric column? (A simple mean wouldn't do). I note a reference to Iqbal (1983) but elaboration would help the reader. Section 3.3: PAR itself is an excellent proxy for cloudiness. Why is precipitation used? Of course it is almost always cloudy when rain is observed but of course more often than not there are clouds but no rain. Page 6 line 19: Was AT-Neu chosen because it is the first alphabetically? This site is in a north-facing mountain valley and there will be shielding of the sun by mountains to the east and west in the early morning and late afternoon.

Figures S1 to S28 is a bit too much information, even for a supplement. For Fig. 2C of course there is a 180 day negative autocorrelation because of solar geometry. It is interesting to see that c and k have somewhat more complicated long-term autocorrelation functions but is there a better way to synthesize this than to create 28 figures in a Supplement? P. 6 L. 22: Too many of these statements are obvious and follow directly from the solar zenith angle and the amount of atmosphere that a beam has to travel through when the sun is not directly overhead. Also, what does this statement mean 'In these sites, climatic seasonality is very weak since the low ACF after removing the astronomical seasonality.' That the statistics of PAR, c, and k are controlled by solar geometry rather than climate? Of course this isn't surprising but it would be interesting to see that proportion of the variables are explained by climate vs. solar geometry, like a variance decomposition. How much of the variability at each site is explained by these two factors and does Koeppen climate classification help explain some of the variability that is not explained by latitude alone? It is still not clear to me what 'bimodal' means. This is a scale-dependent term. More than one peak per day? More than one peak per season? The statement on page 7 line 9 isn't supported directly by a figure and I am still confused as to what the major objective of the manuscript is. P 8: reword: allowing to analyze schematically Does Fig. 4 directed at the notion that some sites have darker clouds than others because of the distributions of c and

k on wet and dry days? Figure 5: was a Bonferroni correction applied to significance values? Also, please do not simultaneously use red and green in the same figure. Also, why are both KS and AD tests used? What advantages do they each have and why not choose just one? Are the values in the boxes p-values and why are they frequently greater than 1? The paragraph after Figure 5 is confusing (p. 11 line 1). I'm not sure what it means: are the data being used to define when seasons begin and end? Figure 6: Please avoid rainbow color schemes (https://eos.org/features/the-end-of-the-rainbow-color-schemes-for-improved-data-graphics). Also, the relationship between k and c is merely PAR_0 / PAR_cda. This figure only shows how much atmosphere there is which of course is greater at high latitudes in winter when the sun is arriving at an angle (no idea what is happening with US-SRM). It is an inefficient way of showing the effects of the solar zenith angle on surface radiation. I cannot emphasize enough how important it is to have very clear subsections when writing a combined Results and Discussion section. The section jumps surprisingly to different topics throughout and is very difficult to follow. Please add subsections at a minimum to help the reader interpret the flow of the argument. Bottom of page 13: I am still not sure what bimodal means in this context and why the analysis is extended to Holdridge life zones. Do some of these ecosystems have expected diurnal or seasonal variability in cloudiness such that grouping the analysis by life zone makes sense? Also, one would expect that a manuscript submitted to Biogeosciences would discuss the importance of the findings to biogeoscience. In this case the role of PAR in controlling photosynthesis is a logical connection. The paper would be stronger if implications for biogeoscience were discussed in more detail. I want to very strongly recommend that the analysis have separate Results and Discussions sections to make it easier to follow and to make the importance of the analysis more clear.

---

## Referee Comment (RC2) · Anonymous Referee #3 · 5 Jun 2020

This was an interesting paper about atmospheric attenuation of photosynthetically active radiation (PAR). The paper addresses the spatiotemporal variability in atmospheric attenuation of PAR by analyzing and characterizing the clearness index and the clear-day index calculated from long-term observational PAR data for near-globally dispersed sites. The paper provides us with the patterns in atmospheric attenuation of PAR that can be expected for various ecosystems according to their position on the Holdridge triangle or their Köppen climate classification. I enjoyed reading about the indices and the spatiotemporal patterns the researchers have found at a large scale, but the reasons for undertaking the research could be expanded upon. Below are some specific comments that I had:

Title: The impression I got from the paper is that it characterizes the site level patterns in atmospheric attenuation that impact how much PAR reaches the ground. The title could be a bit more detailed to include the indices or atmospheric attenuation rather than just "daily PAR".

Abstract: The abstract does not communicate why this research was undertaken. The importance of PAR is briefly described in the introduction, but there is no mention of it in the abstract. A sentence about why we should analyze the variability in atmospheric attenuation of PAR in the beginning and another sentence about why the findings or methods are important in the end could help form a complete abstract.

Introduction: At lines 21 and 22, the authors introduce the indices and mention their wide use by other researchers to "quantify the random nature of atmospheric light attenuation" without references to research. The introduction could be expanded to clarify the purpose of studying the variability in atmospheric attenuation of PAR. Some questions below might help expand the introduction: 1. Which studies used the indices to study the variability of atmospheric attenuation? 2. What did those studies find and how does this current research build on previous studies of atmospheric attenuation? 3. Has the variability in the indices been characterized according to climate in the past? If not, why do the authors believe it is important to characterize the variability in atmospheric attenuation by life zone or climate?

Line 12 on pg 6 mentions that the data was separated into rainy and dry days using precipitation. No precipitation dataset is described in the data section. Adding a description of the source for the precipitation dataset will be helpful.

Line 18 on pg 6 says: "The time series, annual cycle, and autocorrelogram of PAR, c and k were calculated and plotted for each site." Is this referring to PAR0 or PARobs ? It might be helpful to add that the time series, annual cycle, and autocorrelogram were calculated for PAR in the methods section.

Figure 2 and the corresponding supplementary figures show what appears to be a

BGD

 **BGD**

Interactive
comment

confidence interval for the ACF with a dotted line. Which level of confidence does that interval mark?

Technical corrections:

Figure 2 and figures S1 - S28 need legends with a clarification on which PAR measurement is plotted (PAR0 or PARobs).

It is really hard to read the numbers on the figures with the CDF labeled with numbers (figure 5 and figures S57- S84).

Throughout the paper and figure captions, the parentheses come before the variable they describe. For example: "(a-b) c and (c-d) k". It is a bit easier to read if the variable is mentioned first: "c (a-b) and k (c-d)".

At points in the results/discussion, the figures are introduced by describing the figure. For example: " Fig. 4 shows the PDFs (left panel) and the CDFs (right panel) for wet (blue) and dry (red) days of c (a–b) and k (c–d)." (Pg. 8, line 17). This seems redundant. A good descriptive caption for the figure or a complete legend should take care of this and the text in the results/discussion does not need to mention it.

Line 18 on pg. 8 should read: "Figs. S26 to S56 show the results of the 28 sites analyzed."

Regarding lines 11 - 14 on pg. 7: "We classified the pdfs of c and k in three types: Bimodal, Unimodal I (unimodal with low dispersion), and Unimodal II (unimodal with high dispersion). Sites in the extratropical northern hemisphere (except the site in the United States US-Fep) have bimodal distributions; sites in tropics, subtropics, and US-Fpe have Unimodal II distributions; and sites in tropics have Unimodal II distributions." This appears to be in disagreement with figure 3. US-Fpe looks like it has a Unimodal I distribution in figure 3.

If possible, harmonizing the terminology that describes the PDFs between the abstract, results, figures, and conclusion would be helpful. For example, eliminating unimodal I

and II altogether and keeping unimodal low and unimodal high to describe the unimodal PDFs throughout the paper and figures should provide consistency for the reader. I also find unimodal low and unimodal high to be more descriptive.

When talking about the PDFs on pg. 7 and 8: The current organization of paragraphs: Discusses the PDFs' latitudinal variability on pg. 7 - top of pg. 8, then talks about the Köppen classification, and then talks about the Holdridge triangle with mention of latitudinal variability. Consider moving the paragraph about the Köppen classification (lines 3 - 7, pg. 8) before mentioning the Holdridge triangle and latitudinal variability so that the discussion on the latitudinal variability is continuous. An order such as: Introduce the classification of the PDFs, then discuss Köppen classification of site PDFs, and then discuss Holdridge triangle position and latitudinal variability of site PDFs

What does "NEP-WCMC" stand for on pg 3 line 12?

There seems to be some disagreement between the abstract and the conclusion. The abstract says: "Unimodal distributions with high dispersion are concentrated in the moist forest life zone in subtropical and tropical regions and humid province; and unimodal distributions with low dispersion are concentrated in dry forest, very dry forest, and thorn woodland in tropical and subtropical regions between arid and subhumid humidity provinces." The conclusion says: "High latitudes sites exhibit bimodal distributions, arid to sub- humid climates exhibit unimodal distributions with high dispersion, and humid tropical regions exhibit unimodal distributions with low dispersion."

---

## Author Comment (AC2) · 27 Jun 2020

Dear Reviewer 1, We sincerely thank Reviewer 1 (Anonymous Referee #2) for his comprehensive comments, which will be of great help in improving our manuscript. Please see below for a point-by-point response to your comments.

1. Munoz and Ochoa explore patterns of PAR across different latitudes and climate zones. The analysis may be important to the extent that it helps organize and communicate variability in photosynthetically active radiation to the biogeosciences community. At the moment it does not, but I feel that it might. Namely, if the amount of variation in PAR, c, and k explained by solar geometry (obvious) and climate (less obvious) could

be determined I could see how obvious aspects of the manuscript could be placed in the context of information that could be quite enlightening for our understanding of how light reaches the surface across the globe. If the authors can do this I feel that the manuscript could be acceptable for publication; at the moment the findings are largely either obvious or unclear, but the latter can be fixed by restructuring the manuscript and explaining more clearly what was done and its implications. Author response: Yes, there are some obvious results that are not the core of our study and we should state it clearly. Although we analyze PAR, c and k, our main interest is on c and k. Our important (and less obvious) findings are that: a) The PDFs of c and k have the same shape at each site. b) We identified three types of PDF: unimodal with low dispersion (Unimodal I), unimodal with high dispersion (Unimodal II), and bimodal. c) PDFs are unimodal for all the dry life zones studied, bimodal for low rainfall (< ∼1000 mm/yr) humid life zones and unimodal with high dispersion for high rainfall (2000-4000 mm/yr) humid zones. d) There is one Holdridge life-zone class that is a triple point where the three PDF types occur. It is the "Moist forest" with ETp ratio between 0.5 and 1, annual precipitation between 1000 and 2000 mm and humidity province Humid. The variability of PAR is of course explained by latitude and solar geometry, but c and k variability seem to be strongly influenced by the local (maybe regional) climate/life zones. If the Editor agrees, we will write our manuscript more clearly.

2. Regarding "Attenuation of light throughout the atmosphere can be calculated by using an attenuation law (e.g. the Beer–Lambert law), but this requires to know the atmospheric optical depth, which is seldom the case" it is also important to note comprehensive atmospheric modeling that seeks to understand the dynamics of atmospheric transmissivity, reflectivity, and absorptivity as a function of wavelength and layer of the atmosphere. Such models are great but difficult to implement at large scales. Author response: We model atmospheric attenuation as a function of wavelength for a clean and dry atmosphere along the day, and then integrate over the PAR spectral domain and daylength to obtain the daily PAR. We do not model attenuation by clouds and aerosols but expect this attenuation to be quantified by the cloudiness index (c).
3. Page 1 Line 22: light attenuation is not random, it is a function of the physics of the atmosphere. Author response: Yes. What is random is the amount of aerosols and clouds in the atmosphere. We will write it more clearly.

4. Page 1 Line 25: More evidence is needed that this is the case in the form of references. The Introduction as a whole was a bit too brief. Specifically the notion that c and k are stochastic needs to be addressed in more detail. In many regions, clouds are rather predictable like in areas where sea breezes create weather systems that are easy to anticipate. Fog is another atmospheric phenomena that is expected and predictable in certain times and certain reasons. I had coastal California ecosystems in mind when writing that but then noted that this paper was published just today. (https://agupubs.onlinelibrary.wiley.com/doi/abs/10.1029/2020GL088428) Please expand the introduction to discuss the variables that change c and k. Author response: 1) We will discuss in more detail the use daily c and k including the most relevant references (e.g., Utrillas et al. (2018), Ineichen (2016), Engerer & Mills (2014), Tran (2013), Hollands & Suehrcke (2013), Harrouni (2008), Ianetz & Kudish (2008), Polo et al. (2008), Tovar-Pescador (2008), Allen et al. (2006), Assunção et al. (2003), Cañada et al. (2003), Ibáñez et al. (2002), Hansen (1999), Martinez-Lozano et al. (1999), Skartveit & Olseth (1992), Gordon & Hochman (1984), Olseth & Skartveit (1984), Bendt et al. (1981), and Liu & Jordan (1960)). 2) Although cloudiness could be predictable in some regions, our interest is not in a forecasting model. Our objective is to know the statistical behavior of c and k in relation to local climate and life zone to use it as a driver for ecohydrological and biogeochemical stochastic models in energy-limited ecosystems (see Muñoz et al. 2020). We will clarify this point in the manuscript. Allen, R.G., Trezza, R., & Tasumi, M. (2006). Analytical integrated functions for daily solar radiation on slopes. Agricultural and Forest Meteorology, 139, 55–73. https://doi.org/10.1016/j.agrformet.2006.05.012 Assunção, H. F., Escobedo, J. F., & Oliveira, A. P. (2003). Modelling frequency distributions of 5 minute-averaged solar radiation indexes using Beta probability functions. Theoretical and Applied Climatology, 75(3–4), 213–224. https://doi.org/10.1007/s00704-003-0733-9 Bendt, P.,

[Figure]

Collares-Pereira, M., & Rabl, A. (1981). The frequency distribution of daily insolation values. Solar Energy, 27, 1–5. Cañada, J., Pedros, G., & Bosca, J.V. (2003). Relationships between UV (0.290–0.385 $\mu$m) and broad band solar radiation hourly values in Valencia and Córdoba, Spain. Energy, 28(3), 199–217. https://doi.org/10.1016/S0360-5442(02)00111-1 Engerer, N.A., & Mills, F.P. (2014). KPV: A clear-sky index for photovoltaics. Solar Energy, 105, 679–693. https://doi.org/10.1016/j.solener.2014.04.019 Gordon, J.M., & Hochman, M. (1984). On the random nature of solar radiation. Solar Energy, 32(3), 337–342. https://doi.org/10.1016/0038-092X(84)90276-7 Hansen, J.W. (1999). Stochastic daily solar irradiance for biological modeling applications. Agricultural and Forest Meteorology, 94(1), 53–63. https://doi.org/10.1016/S0168-1923(99)00003-9 Harrouni, S. (2008). Fractal Classification of Typical Meteorological Days from Global Solar Irradiance: Application to Five Sites of Different Climates. In V. Badescu (Ed.), Modeling Solar Radiation at the Earth's Surface (pp. 29–55). Retrieved from http://link.springer.com/chapter/10.1007/978-3-540-77455-6_2%0Ahttp://files/1185/Harrouni - 2008 - Fractal Classification of Typical Meteorological D.pdf%0Ahttp://files/1192/10.html Hollands, K.G.T., & Suehrcke, H. (2013). A three-state model for the probability distribution of instantaneous solar radiation, with applications. Solar Energy, 96, 103–112. https://doi.org/10.1016/j.solener.2013.07.007 Ianetz, A., & Kudish, A. (2008). A method for determining the solar global and defining the diffuse and beam irradiation on a clear day. In V. Badescu (Ed.), Modeling Solar Radiation at the Earth's Surface: Recent Advances (pp. 93–113). https://doi.org/10.1007/978-3-540-77455-6_4 Ibánez, M., Rosell, J. I., & Beckman, W. A. (2003). A bi-variable probability density function for the daily clearness index. Solar Energy, 75(1), 73–80. https://doi.org/10.1016/S0038-092X(03)00123-3 Ineichen, P. (2016). Validation of models that estimate the clear sky global and beam solar irradiance. Solar Energy, 132, 332–344. https://doi.org/10.1016/j.solener.2016.03.017 Liu, B.Y.H., & Jordan, R.C. (1960). The interrelationship and characteristic distribution of direct, diffuse and total solar radiation. Solar Energy, 4(3), 1–19. https://doi.org/10.1016/0038-092X(60)90062-1 Martinez-Lozano, J. A., Tena, F., & Utrillas, M. P. (1999). Ratio of UV

to global broad band irradiation in Valencia, Spain. International Journal of Climatology, 19(8), 903–911. https://doi.org/10.1002/(SICI)1097-0088(19990630)19:8<903::AID-JOC400>3.0.CO;2-N Muñoz, E., Ochoa, A., Poveda, G., & Rodríguez-Iturbe, I. (2020). Probabilistic soil moisture dynamics of water- and energy-limited ecosystems. Earth-ArXiv. https://doi.org/10.31223/osf.io/au4tb Olseth, J.A., & Skartveit, A. (1984). A probability density function for daily insolation within the temperate storm belts. Solar Energy, 33(6), 533–542. https://doi.org/10.1016/0038-092X(84)90008-2 Polo, J., Zarzalejo, L. F., & Ramírez, L. (2008). Solar Radiation Derived from Satellite Images. In V. Badescu (Ed.), Modeling Solar Radiation at the Earth's Surface (pp. 449–461). Springer-Verlag Berlin Heidelberg. Skartveit, A., & Olseth, J. A. (1992). The probability density and autocorrelation of short-term global and beam irradiance. Solar Energy, 49(6), 477–487. https://doi.org/10.1016/0038-092X(92)90155-4 Tovar-Pescador, J. (2008). Modelling the statistical properties of solar radiation and proposal of a technique based on Boltzmann statistics. In V. Badescu (Ed.), Modeling Solar Radiation at the Earth's Surface: Recent Advances (pp. 55–91). https://doi.org/10.1007/978-3-540-77455-6_3 Tran, V. L. (2013). Stochastic models of solar radiation processes. Université d'Orléans. Utrillas, M.P., Marín, M.J., Esteve, A.R., Salazar, G., Suárez, H., Gandía, S., & Martínez-Lozano, J.A. (2018). Relationship between erythemal UV and broadband solar irradiation at high altitude in Northwestern Argentina. Energy, 162(August), 136–147. https://doi.org/10.1016/j.energy.2018.08.021

5. Note also that PAR and the shortwave bands overlap, but incompletely. If you are studying PAR, simply explain why and what the important differences are. Author response: We are interested in characterizing c and k for PAR to use them as inputs in stochastic ecohydrological and biogeochemical modeling. We will incorporate this context in the manuscript.

6. Adding the Holdridge/Koppen zones to Table 1 would be an improvement. Author response: It will be done.

7. Why only these 28 sites? There are a number of high-latitude sites with long-term

consistent PPFD, for example. Author response: We initially took information from 292 sites from FLUXNET. The initial criteria to select the sites to analyze were: records with a length greater than 5 years (76 sites) and with complete time series of precipitation data (59). Then, we visually selected the sites with suitable information and were not very close to each other. As the sites obtained were mainly located in the northern hemisphere, we were laxer selecting sites in the southern hemisphere. For those, we chose the sites with longer time series, without apparent errors and precipitation information.

8. Page 4 Line 6: this is true but requires elaboration: 'troublesome when using the Beer–Lambert law'. It is certainly troublesome if the atmosphere is considered to be one layer because atmospheric attenuation will vary dramatically by layer over time, but a layer-by-layer implementation of the Beer-Lambert Law over short time scales may be quite accurate but difficult to implement. Author response: We agree with your comment. We will expand this idea in the manuscript.

9. After equation 4: 'transmittance due to molecular absorbers of': please note that this is for the clean and dry atmosphere for this particular calculation ('cda') so that people realize why aerosols and other nonmolecular absorbers (and reflecters) are excluded. Author response: We will clarify it.

10. Why is forward / back scattering of 0.5 assumed? Please elaborate in the text. Author response: This parameter is very difficult to estimate. We used 0.5 arbitrarily and expect that the uncertainty and variability of this parameter be expressed in the clarity index (c).

11. In equation 5, how much do higher-order reflectances typically contribute? It might not be minor, I'm not sure. Author response: It may be important in the presence of a high surface albedo. Indeed, we got some c>1 values during winter in sites with seasonal snow. We report this effect in line 20 of page 6. When c>1 the contribution of high-order reflectances is clear, but it could go unnoticed when c<1. The c index

represents uncertainty from several sources, and that is why it has different names as "cloudiness index", "clear-sky index" and "clarity index". In our approach, c should also include the uncertainty of albedo and high order reflection.

12. In equation 6, how is ozone derived? Is it weighted for its distribution throughout the atmospheric column? (A simple mean wouldn't do). I note a reference to Iqbal (1983) but elaboration would help the reader. Author response: We use the seasonal variation of atmospheric ozone from Iqbal (1983, p.89) that gives the total amount of ozone in a vertical column of air for several latitudes. We interpolate for the latitude and doy of interest.

13. Section 3.3: PAR itself is an excellent proxy for cloudiness. Why is precipitation used? Of course it is almost always cloudy when rain is observed but of course more often than not there are clouds but no rain. Author response: This is a good point and deserves an interesting statistical approach, but it is out of our scope for now. We use rainfall because we have in mind the connection between our statistical characterization of c and k and the ecohydrological and biogeochemical models of Rodríguez-Iturbe and collaborators. That family of models study the stochastic balance equation (for water, carbon, nitrogen) of the soil forced by a Poisson process rainfall model with parameter lambda. In a future research, we plan to study the joint PDF of daily rainfall (occurrence and amount) and PAR. We will incorporate this context in the introduction and conclusions of the manuscript.

14. Page 6 line 19: Was AT-Neu chosen because it is the first alphabetically? This site is in a north-facing mountain valley and there will be shielding of the sun by mountains to the east and west in the early morning and late afternoon. Author response: AT-Neu was selected because it has long and high-quality records. You are right about the local orography at this site. That local trait, as well as Holdridge life zone or Köppen climate zone, must be reflected in the PDF of c and k.

15. Figures S1 to S28 is a bit too much information, even for a supplement. For Fig.

2C of course there is a 180 day negative autocorrelation because of solar geometry. It is interesting to see that c and k have somewhat more complicated long-term autocorrelation functions but is there a better way to synthesize this than to create 28 figures in a Supplement? Author response: We agree that panel (c) inf Figures S1 to S28 is not very informative. We think, however, that panels (a), (b) and (d) to (i) are a good support of the analysis. Removing panel (c) does not reduce too much the amount of information in the Supplement. We would appreciate more suggestions about this from you, the other reviewers and the editor.

16. P. 6 L. 22: Too many of these statements are obvious and follow directly from the solar zenith angle and the amount of atmosphere that a beam has to travel through when the sun is not directly overhead. Also, what does this statement mean 'In these sites, climatic seasonality is very weak since the low ACF after removing the astronomical seasonality.' That the statistics of PAR, c, and k are controlled by solar geometry rather than climate? Of course, this isn't surprising but it would be interesting to see that proportion of the variables are explained by climate vs. solar geometry, like a variance decomposition. How much of the variability at each site is explained by these two factors and does Koeppen climate classification help explain some of the variability that is not explained by latitude alone? Author response: We agree that some statements are obvious for PAR and they should be removed/rewritten, but they are not so obvious for c and k. The analysis of ACF is not clear and we will rewrite it. We analyzed the effect of the local climate on the shape of the PDF of c and k by visual inspection. However, it would be very enlightening to quantify it by a variance decomposition, as you suggest. We can do it if the editor agrees.

17. It is still not clear to me what 'bimodal' means. This is a scale-dependent term. More than one peak per day? More than one peak per season? Author response: "Bimodal" and "unimodal" are terms used in tropical hydrology to describe the shape of the annual regime of hydrologic variables. Unimodal refers to one mode and bimodal to two modes (e.g. two rainfall seasons, two growing seasons (Knoben et al. (2019), Urrea et al. (2019), Herrmann & Mohr (2011)). We will incorporate this explanation in the manuscript. Herrmann, S.M., & Mohr, K.I. (2011). A Continental-Scale Classification of Rainfall Seasonality Regimes in Africa Based on Gridded Precipitation and Land Surface Temperature Products. Journal of Applied Meteorology and Climatology, 50(12), 2504–2513. https://doi.org/10.1175/JAMC-D-11-024.1 Knoben, W.J.M., Woods, R.A., & Freer, J.E. (2019). Global bimodal precipitation seasonality: A systematic overview. International Journal of Climatology, 39(1), 558–567. https://doi.org/10.1002/joc.5786 Urrea, V., Ochoa, A., & Mesa, O. (2019). Seasonality of Rainfall in Colombia. Water Resources Research, 55(5), 4149–4162. https://doi.org/10.1029/2018WR023316

18. The statement on page 7 line 9 isn't supported directly by a figure and I am still confused as to what the major objective of the manuscript is. Author response: This paragraph is not clear and we will rewrite it. The main objective of this study is to characterize the PDF of c and k according to the local climate/life zone. Our afterward interest is to use this characterization as an input to stochastic ecohydrological and biogeochemical models (e.g. Ridolfi et al. (2003), Manzoni et al (2004), Botter et al. (2018), Runyan and D'Odorico (2019), Manzoni et al. (2020), Muñoz et al. (2020)), but it will be useful also in other fields such as agronomy, geotechnics and natural hazard analysis. Botter, G., Daly, E., Porporato, A., Rodríguez-Iturbe, I., & Rinaldo, A. (2008). Probabilistic dynamics of soil nitrate: Coupling of ecohydrological and biogeochemical processes. Water Resources Research, 44(3), n/a-n/a. https://doi.org/10.1029/2007WR006108 Manzoni, S., Porporato, A., D'Odorico, P., Laio, F., & Rodriguez-Iturbe, I. (2004). Soil nutrient cycles as a nonlinear dynamical system. Nonlinear Processes in Geophysics, 11(5/6), 589–598. https://doi.org/10.5194/npg-11-589-2004 Manzoni, S., Chakrawal, A., Fischer, T., Schimel, J. P., Porporato, A., & Vico, G. (2020). Rainfall intensification increases the contribution of rewetting pulses to soil respiration. Biogeosciences Discussions, 1–25. https://doi.org/10.5194/bg-2020-95 Muñoz, E., Ochoa, A., Poveda, G., & Rodríguez-Iturbe, I. (2020). Probabilistic soil moisture dynamics of water- and energy-limited ecosystems. EarthArXiv. https://doi.org/10.31223/osf.io/au4tb Ridolfi, L., D'Odorico, P., Porporato, A., & Rodriguez-Iturbe, I. (2003). The influence of stochastic soil moisture dynamics on gaseous emissions of NO, N2O, and N2. Hydrological Sciences Journal, 48(5), 781–798. https://doi.org/10.1623/hysj.48.5.781.51451 Runyan, C. W., & D'Odorico, P. (2019). Modeling of Phosphorus Dynamics in Dryland Ecosystems. In Dryland Ecohydrology (pp. 309–333). Springer International Publishing. https://doi.org/10.1007/978-3-030-23269-6_12

19. P 8: reword: allowing to analyze schematically Does Fig. 4 directed at the notion that some sites have darker clouds than others because of the distributions of c and k on wet and dry days? Author response: We will improve writing. All panels in Fig. 4 are for AT-Neu, top panels for c index and bottom panels for k index. The difference between top panels and bottom panels comes from the clean and dry atmosphere and is an obvious result as you pointed out before. We use Fig. 4 to show that the bimodal PDF of c and k is the superposition of two unimodal PDFs, ones for dry and one for wet days. Fig.3 suggests that sites within the same life zone have c and k PDFs with similar shapes. Moreover, the shape of the PDFs is similar between neighboring life zones. Our interpretation of this pattern is that the life zone is the result of adaptation to the local climate, although at large spatial and time scales the climate could be a result of the ecosystem. We will clarify this point in the manuscript.

20. Figure 5: was a Bonferroni correction applied to significance values? Also, please do not simultaneously use red and green in the same figure. Also, why are both KS and AD tests used? What advantages do they each have and why not choose just one? Are the values in the boxes p-values and why are they frequently greater than 1? Author response: No, we didn't because we are not doing several tests at a time. We test all combinations of pairs of months. We can change the green/red in Fig. 5. We used the two tests because AD is more sensitive to the tails of the distribution while KS to the center of the distribution. We will discuss it in the manuscript, confronting the result with both tests. The p-values in the boxes are greater than 1 because they are multiplied by 100 to show more decimals using less space. It will be clarified in the legend of the figure.

21. The paragraph after Figure 5 is confusing (p. 11 line 1). I'm not sure what it means: are the data being used to define when seasons begin and end? Author response: We are comparing the monthly PDFs of c for the dry-days and wet-days samples to inspect for the existence (and duration) of c and k seasons (i.e. groups of months where c and k have the same PDF). We will write it more clearly.

22. Figure 6: Please avoid rainbow color schemes (https://eos.org/features/the-end-of-therainbow-color-schemes-for-improved-data-graphics). Also, the relationship between k and c is merely PAR_0 / PAR_cda. This figure only shows how much atmosphere there is which of course is greater at high latitudes in winter when the sun is arriving at an angle (no idea what is happening with US-SRM). It is an inefficient way of showing the effects of the solar zenith angle on surface radiation. Author response: Thank you for pointing this out, we will change it to a two-color band. Yes, it is PAR0/PARcda and changes with latitude in an obvious way. However, this effect is asymmetric between the northern and southern hemispheres. This asymmetry is also obvious and is explained by the eccentricity in the Earth's orbit. We will remove this figure and this analysis.

23. I cannot emphasize enough how important it is to have very clear subsections when writing a combined Results and Discussion section. The section jumps surprisingly to different topics throughout and is very difficult to follow. Please add subsections at a minimum to help the reader interpret the flow of the argument. I want to very strongly recommend that the analysis have separate Results and Discussions sections to make it easier to follow and to make the importance of the analysis more clear Author response: We will separate Results and Discussion in two sections and will add the necessary subsections to have a more clear structure.

24. Bottom of page 13: I am still not sure what bimodal means in this context and why the analysis is extended to Holdridge life zones. Do some of these ecosystems have expected diurnal or seasonal variability in cloudiness such that grouping the analysis by life zone makes sense? Author response: Please see our response to comment 17

on "unimodal" and "bimodal" terms. Since latitude is not enough to explain the shape of the PDF of c and k (e.g. USA-FPe, 48°N, is unimodal while CH-Oe1, 47.3°N, is bimodal), we extended our analysis to the Holdridge life zones looking for bioclimatic traits in c and k statistics. We use daily data and do not analyze any intra-diurnal variability. We will revise the manuscript to avoid this misinterpretation. On ecosystems and cloudiness: Yes, we expect that ecosystems in the same life zone have similar cloudiness regimes, not necessarily because they force atmospheric processes (these could happen at large spatial and temporal scales) but because they are the result of adaptation to the local climate.

25. Also, one would expect that a manuscript submitted to Biogeosciences would discuss the importance of the findings to biogeosciences. In this case, the role of PAR in controlling photosynthesis is a logical connection. The paper would be stronger if implications for biogeoscience were discussed in more detail. Author response: Our study has important findings to biogeoscience and we will highlight them. Please see details in our response to your comment 18.

Estefanía Muñoz Andrés Ochoa .

---

## Author Comment (AC3) · 27 Jun 2020

Dear Referee 3,

We appreciate the time and effort to provide feedback on our manuscript and are grateful for your valuable suggestions. We will incorporate these suggestions in the final version of the manuscript. Please see below for a point-by-point response to your comments.

1. This was an interesting paper about the atmospheric attenuation of photosynthetically active radiation (PAR). The paper addresses the spatiotemporal variability in at-

[Figure]

mospheric attenuation of PAR by analyzing and characterizing the clearness index and the clearday index calculated from long-term observational PAR data for near-globally dispersed sites. The paper provides us with the patterns in atmospheric attenuation of PAR that can be expected for various ecosystems according to their position on the Holdridge triangle or their Köppen climate classification. I enjoyed reading about the indices and the spatiotemporal patterns the researchers have found on a large scale, but the reasons for undertaking the research could be expanded upon. Author response: Thank you for your comment. What led us to undertake this research was the need for a probability model of daily radiation to investigate the stochastic dynamics of soil water, nitrogen and carbon contents in energy-limited ecosystems, just as it has been done for water-limited ecosystems (e.g. Ridolfi et al. (2003), Manzoni et al (2004), Botter et al. (2018), Runyan and D'Odorico (2019), Manzoni et al. (2020)). Our ultimate goal is to extend the ecohydrological model of Rodríguez-Iturbe and coauthors from water-limited to energy-limited ecosystems. We are currently working on it (see Muñoz et al. (2020)). We will expand the introduction with this context. Botter, G., Daly, E., Porporato, A., Rodríguez-Iturbe, I., & Rinaldo, A. (2008). Probabilistic dynamics of soil nitrate: Coupling of ecohydrological and biogeochemical processes. Water Resources Research, 44(3), n/a-n/a. https://doi.org/10.1029/2007WR006108 Manzoni, S., Porporato, A., D'Odorico, P., Laio, F., & Rodriguez-Iturbe, I. (2004). Soil nutrient cycles as a nonlinear dynamical system. Nonlinear Processes in Geophysics, 11(5/6), 589–598. https://doi.org/10.5194/npg-11-589-2004 Manzoni, S., Chakrawal, A., Fischer, T., Schimel, J. P., Porporato, A., & Vico, G. (2020). Rainfall intensification increases the contribution of rewetting pulses to soil respiration. Biogeosciences Discussions, 1–25. https://doi.org/10.5194/bg-2020-95 Muñoz, E., Ochoa, A., Poveda, G., & Rodríguez-Iturbe, I. (2020). Probabilistic soil moisture dynamics of water- and energy-limited ecosystems. EarthArXiv. https://doi.org/10.31223/osf.io/au4tb Ridolfi, L., D'Odorico, P., Porporato, A., & Rodriguez-Iturbe, I. (2003). The influence of stochastic soil moisture dynamics on gaseous emissions of NO, N2O, and N2. Hydrological Sciences Journal, 48(5), 781–798. https://doi.org/10.1623/hysj.48.5.781.51451 Runyan, C. W., & D'Odorico, P. (2019). Modeling of Phosphorus Dynamics in Dryland Ecosystems. In Dryland Ecohydrology (pp. 309–333). Springer International Publishing. https://doi.org/10.1007/978-3-030-23269-6_12

2. Title: The impression I got from the paper is that it characterizes the site level patterns in atmospheric attenuation that impact how much PAR reaches the ground. The title could be a bit more detailed to include the indices or atmospheric attenuation rather than just "daily PAR". Author response: We will change to "Bioclimatic traits in atmospheric attenuation of daily photosynthetically active radiation".

3. Abstract: The abstract does not communicate why this research was undertaken. The importance of PAR is briefly described in the introduction, but there is no mention of it in the abstract. A sentence about why we should analyze the variability in atmospheric attenuation of PAR in the beginning and another sentence about why the findings or methods are important in the end could help form a complete abstract. Author response: the abstract will be completed according to your suggestions. We will mention that PAR is the main source of energy in photosynthesis and evapotranspiration and that the amount of energy reaching the Earth's surface at a given time and space depends on part of the atmospheric attenuation. Besides, we will add that the results found here show that the stochastic component of PAR radiation at a site can be associated with its Holdridge life zones and Köppen climate.

4. Introduction: At lines 21 and 22, the authors introduce the indices and mention their wide use by other researchers to "quantify the random nature of atmospheric light attenuation" without references to research. The introduction could be expanded to clarify the purpose of studying the variability in atmospheric attenuation of PAR. Some questions below might help expand the introduction: 1. Which studies used the indices to study the variability of atmospheric attenuation? 2. What did those studies find and how does this current research build on previous studies of atmospheric attenuation? 3. Has the variability in the indices been characterized according to climate in the past? If not, why do the authors believe it is important to characterize the variability in atmospheric attenuation by life zone or climate? Author response: Thank you for your suggestions. We will complete the introduction with references to previous work about the indices (e.g., Engerer & Mills (2014), Tran (2013), Hollands & Suehrcke (2013), Harrouni (2008), Ianetz & Kudish (2008), Polo et al. (2008), Allen et al. (2006), Hansen (1999), Skartveit & Olseth (1992), Gordon & Hochman (1984), Olseth & Skartveit (1984), Bendt et al. (1981), and Liu & Jordan (1960)). We will also respond to the three questions suggested based on the results of the mentioned (and others) works.

Allen, R. G., Trezza, R., & Tasumi, M. (2006). Analytical integrated functions for daily solar radiation on slopes. Agricultural and Forest Meteorology, 139, 55–73. https://doi.org/10.1016/j.agrformet.2006.05.012 Bendt, P., Collares-Pereira, M., & Rabl, A. (1981). The frequency distribution of daily insolation values. Solar Energy, 27, 1–5. Engerer, N. A., & Mills, F. P. (2014). KPV: A clear-sky index for photo-voltaics. Solar Energy, 105, 679–693. https://doi.org/10.1016/j.solener.2014.04.019 Gordon, J. M., & Hochman, M. (1984). On the random nature of solar radiation. Solar Energy, 32(3), 337–342. https://doi.org/10.1016/0038-092X(84)90276-7 Hansen, J. W. (1999). Stochastic daily solar irradiance for biological modeling applications. Agricultural and Forest Meteorology, 94(1), 53–63. https://doi.org/10.1016/S0168-1923(99)00003-9 Harrouni, S. (2008). Fractal Classi cation of Typical Meteorological Days from Global Solar Irradiance: Application to Five Sites of Different Climates. In V. Badescu (Ed.), Modeling Solar Radiation at the Earth's Surface (pp. 29–55). Retrieved from http://link.springer.com/chapter/10.1007/978-3-540-77455-6_2%0Ahttp://files/1185/Harrouni - 2008 - Fractal Classification of Typical Meteorological D.pdf%0Ahttp://files/1192/10.html Hollands, K. G. T., & Suehrcke, H. (2013). A three-state model for the probability distribution of instantaneous solar radiation, with applications. Solar Energy, 96, 103–112. https://doi.org/10.1016/j.solener.2013.07.007 Ianetz, A., & Kudish, A. (2008). A method for determining the solar global and defining the diffuse and beam irradiation on a clear day. In V. Badescu (Ed.), Modeling Solar Radiation at the Earth's Surface: Recent Advances (pp. 93–113).

https://doi.org/10.1007/978-3-540-77455-6_4 Liu, B. Y. H., & Jordan, R. C. (1960). The interrelationship and characteristic distribution of direct, diffuse and total solar radiation. Solar Energy, 4(3), 1–19. https://doi.org/10.1016/0038-092X(60)90062-1 Olseth, J. A., & Skartveit, A. (1984). A probability density function for daily insolation within the temperate storm belts. Solar Energy, 33(6), 533–542. https://doi.org/10.1016/0038-092X(84)90008-2 Polo, J., Zarzalejo, L. F., & Ramírez, L. (2008). Solar Radiation Derived from Satellite Images. In V. Badescu (Ed.), Modeling Solar Radiation at the Earth's Surface (pp. 449–461). Springer-Verlag Berlin Heidelberg. Skartveit, A., & Olseth, J. A. (1992). The probability density and autocorrelation of short-term global and beam irradiance. Solar Energy, 49(6), 477–487. https://doi.org/10.1016/0038-092X(92)90155-4 Tran, V. L. (2013). Stochastic models of solar radiation processes. Université d'Orléans.

5. Line 12 on pg 6 mentions that the data was separated into rainy and dry days using precipitation. No precipitation dataset is described in the data section. Adding a description of the source for the precipitation dataset will be helpful. Author response: we got rainfall data from the same database of PAR (FLUXNET). This will be clarified in the text.

6. Line 18 on pg 6 says: "The time series, annual cycle, and autocorrelogram of PAR, c and k were calculated and plotted for each site." Is this referring to PAR0 or PARobs? Author response: Time series (panel a) and annual cycles (panel b) are shown for PARobs (observed), PAR0 (no atmosphere) and PARcda (clean and dry atmosphere), while the ACF (panel c) shows only PARobs. This will be explained in the text.

7. It might be helpful to add that the time series, annual cycle, and autocorrelogram were calculated for PAR in the methods section. Author response: It will be added in Section 3.3 (Statistical properties of k and c).

8. Figure 2 and the corresponding supplementary figures show what appears to be a confidence interval for the ACF with a dotted line. Which level of confidence does

that interval mark? Author response: It refers to the 95% confidence interval. It will be stated in the figures.

9. Figure 2 and figures S1 - S28 need legends with a clarification on which PAR measurement is plotted (PAR0 or PARobs). Author response: black solid lines in Figure 2(a,b) indicate the PAR for no-atmosphere and thick green solid lines indicate the modeled global radiation. The thin green solid line in Fig. 2(a) is the time series of the observed PAR and the dots in Fig. 2(b) are the mean of PARobs of each day of the year during the record. We will explain this in the legends or captions of Fig. 2 and Figs. S1-S28.

10. It is really hard to read the numbers on the figures with the CDF labeled with numbers (figure 5 and figures S57- S84). Author response: We will use symbols instead of numbers.

11. Throughout the paper and figure captions, the parentheses come before the variable they describe. For example: "(a-b) c and (c-d) k". It is a bit easier to read if the variable is mentioned first: "c (a-b) and k (c-d)". Author response: We will change this.

12. At points in the results/discussion, the figures are introduced by describing the figure. For example: " Fig. 4 shows the PDFs (left panel) and the CDFs (right panel) for wet (blue) and dry (red) days of c (a–b) and k (c–d)." (Pg. 8, line 17). This seems redundant. A good descriptive caption for the figure or a complete legend should take care of this and the text in the results/discussion does not need to mention it. Author response: We will correct the legends and separate the Results and Discussion sections.

13. Line 18 on pg. 8 should read: "Figs. S26 to S56 show the results of the 28 sites analyzed." Author response: This will be corrected.

14. Regarding lines 11 - 14 on pg. 7: "We classified the pdfs of c and k in three types: Bimodal, Unimodal I (unimodal with low dispersion), and Unimodal II (unimodal

with high dispersion). Sites in the extratropical northern hemisphere (except the site in the United States US-Fep) have bimodal distributions; sites in tropics, subtropics, and USFpe have Unimodal II distributions; and sites in tropics have Unimodal II distributions." This appears to be in disagreement with figure 3. US-Fpe looks like it has a Unimodal I distribution in figure 3. Author response: Thank you for pointing this out. The paragraph should be rewritten as: We classified the pdfs of c and k in three types: Bimodal, Unimodal I (unimodal with high dispersion), and Unimodal II (unimodal with low dispersion). Sites in the extratropical northern hemisphere (except the site in the United States US-Fep) have bimodal distributions; sites in tropics, subtropics, and USFpe have Unimodal I distributions; and sites in tropics have Unimodal II distributions.

15. If possible, harmonizing the terminology that describes the PDFs between the abstract, results, figures, and conclusion would be helpful. For example, eliminating unimodal I and II altogether and keeping unimodal low and unimodal high to describe the unimodal PDFs throughout the paper and figures should provide consistency for the reader. I also find unimodal low and unimodal high to be more descriptive. Author response: We will revise this issue in the whole manuscript.

16. When talking about the PDFs on pg. 7 and 8: The current organization of paragraphs: Discusses the PDFs' latitudinal variability on pg. 7 - top of pg. 8, then talks about the Köppen classification, and then talks about the Holdridge triangle with mention of latitudinal variability. Consider moving the paragraph about the Köppen classification (lines 3 - 7, pg. 8) before mentioning the Holdridge triangle and latitudinal variability so that the discussion on the latitudinal variability is continuous. An order such as: Introduce the classification of the PDFs, then discuss Köppen classification of site PDFs, and then discuss Holdridge triangle position and latitudinal variability of site PDF. Author response: We think this is an excellent suggestion, we will consider it in the corrected manuscript.

17. What does "NEP-WCMC" stand for on pg 3 line 12? Author response: It is a typing error, the correct is UNEP-WCMC (https://www.unep-wcmc.org/resources-anddata/holdridges-life-zones). Also, it lacks the complete reference. We will fix it.

18. There seems to be some disagreement between the abstract and the conclusion. The abstract says: "Unimodal distributions with high dispersion are concentrated in the moist forest life zone in subtropical and tropical regions and humid province; and unimodal distributions with low dispersion are concentrated in dry forest, very dry forest, and thorn woodland in tropical and subtropical regions between arid and subhumid humidity provinces." The conclusion says: "High latitudes sites exhibit bimodal distributions, arid to sub-humid climates exhibit unimodal distributions with high dispersion, and humid tropical regions exhibit unimodal distributions with low dispersion." Author response: The appropriate remark is that of the abstract. We will improve the conclusions, especially in relation to the humidity provinces.

Estefanía Muñoz Andrés Ochoa

---

## Author Response (AR1)

Dear Paul C. Stoy
Editor Biogeosciences

We acknowledge the opportunity to submit a revised draft of our manuscript. We have implemented most of the suggestions made by the referees. Please see below a point by point response to the referees concerns and the changes made as a consequence of these comments.

Sincerely,
Estefanía Muñoz and Andrés Ochoa

**Referee 2**

1. Munoz and Ochoa explore patterns of PAR across different latitudes and climate zones. The analysis may be important to the extent that it helps organize and communicate variability in photosynthetically active radiation to the biogeosciences community. At the moment it does not, but I feel that it might. Namely, if the amount of variation in PAR, c, and k explained by solar geometry (obvious) and climate (less obvious) could be determined I could see how obvious aspects of the manuscript could be placed in the context of information that could be quite enlightening for our understanding of how light reaches the surface across the globe. If the authors can do this I feel that the manuscript could be acceptable for publication; at the moment the findings are largely either obvious or unclear, but the latter can be fixed by restructuring the manuscript and explaining more clearly what was done and its implications.

   Author response: We expanded our analysis to also include shortwave radiation. We have now a total of 37 sites. We made figures 3b and 3c to visualize the relation of PDF types with global climate zones and Köppen climate types. Although we analyze radiation, c and k, our main interest is on c and k, not radiation per se. Our important (and less obvious) findings are that:
   a) The PDFs of c and k have the same shape at each site.
   b) We identified three types of PDF: unimodal with low dispersion (ULD), unimodal with high dispersion (UHD), and bimodal (B).
   c) Regionalisation of PDF types can be made by latitude, global climate zones, Köppen climate types, and Holdridge life zones are useful for regionalisation of c and k PDF types. The weakest relation was found for latitude and the stronger for Holdridge life zones (Figs. 3b, 3c and 4).

2. Regarding "Attenuation of light throughout the atmosphere can be calculated by using an attenuation law (e.g. the Beer–Lambert law), but this requires to know the atmospheric optical depth, which is seldom the case" it is also important to note comprehensive atmospheric modeling that seeks to understand the dynamics of atmospheric transmissivity, reflectivity, and absorptivity as a function of wavelength and layer of the atmosphere. Such models are great but difficult to implement at large scales.

   Author response: We wrote the introduction in a more clear form.

3. Page 1 Line 22: light attenuation is not random, it is a function of the physics of the atmosphere.

Author response: Yes. What is random is the quantity of aerosols and clouds in the atmosphere, not the physics of attenuation. We rewrote the introduction avoiding this misunderstanding.

4. Page 1 Line 25: More evidence is needed that this is the case in the form of references. The Introduction as a whole was a bit too brief. Specifically the notion that c and k are stochastic needs to be addressed in more detail. In many regions, clouds are rather predictable like in areas where sea breezes create weather systems that are easy to anticipate. Fog is another atmospheric phenomenon that is expected and predictable in certain times and certain reasons. I had coastal California ecosystems in mind when writing that but then noted that this paper was published just today.
(https://agupubs.onlinelibrary.wiley.com/doi/abs/10.1029/2020GL088428) Please expand the introduction to discuss the variables that change c and k.

Author response: We enriched the introduction with references reinforcing those statements. We agree on the different tastes of randomness that could occur as a consequence of local/regional climate. We wrote it in a more clear form in the introduction and added a forward reference to the discussion section.

5. Note also that PAR and the shortwave bands overlap, but incompletely. If you are studying PAR, simply explain why and what the important differences are.

Author response: We expanded our analysis to also include SW radiation.

6. Adding the Holdridge/Koppen zones to Table 1 would be an improvement.

Author response: We included Table 2 with that information.

7. Why only these 28 sites? There are a number of high-latitude sites with long-term consistent PPFD, for example.

Author response: These sites were selected from an initial set of more than 200 sites after filtering by several criteria as record length, data quality and spatial coverage of the whole group. We extended our analysis to shortwave radiation. This increased the number of sites to 37. We included this description in the data section.

8. Page 4 Line 6: this is true but requires elaboration: 'troublesome when using the Beer–Lambert law'. It is certainly troublesome if the atmosphere is considered to be one layer because atmospheric attenuation will vary dramatically by layer over time, but a layer-by-layer implementation of the Beer-Lambert Law over short time scales may be quite accurate but difficult to implement.

Author response: We included this explanation in the Methods section.

9. After equation 4: 'transmittance due to molecular absorbers of': please note that this is for the clean and dry atmosphere for this particular calculation ('cda') so that people realize why aerosols and other nonmolecular absorbers (and reflecters) are excluded.

Author response: We clarified it in the text.

10. Why is forward / back scattering of 0.5 assumed? Please elaborate in the text.

Author response: Author response: We used 0.5 arbitrarily and expect that the uncertainty and variability of this parameter be expressed in the clarity index (c). We specified this in the text.

11. In equation 5, how much do higher-order reflectances typically contribute? It might not be minor, I'm not sure.

Although higher reflectances could bring about some subestimation of H, especially during snow-cover periods, we think it is not a critical issue for the sake of this study. Uncertainty caused by these two assumptions will be included in the statistical properties of c.

12. In equation 6, how is ozone derived? Is it weighted for its distribution throughout the atmospheric column? (A simple mean wouldn't do). I note a reference to Iqbal (1983) but elaboration would help the reader.

Author response: We use the seasonal variation of atmospheric ozone from Iqbal (1983, p.89) that gives the total amount of ozone in a vertical column of air for several latitudes. We interpolate for the latitude and doy of interest. We rewrote this paragraph in a more clear form.

13. Section 3.3: PAR itself is an excellent proxy for cloudiness. Why is precipitation used? Of course it is almost always cloudy when rain is observed but of course more often than not there are clouds but no rain.

Author response: This is an interesting point deserves an interesting statistical approach, but it is out of our scope in this article. We use rainfall because we have in mind the connection between our statistical characterization of c and k and the ecohydrological and biogeochemical models of Rodríguez-Iturbe and collaborators. That family of models study the stochastic balance equation (for water, carbon, nitrogen) of the soil forced by a Poisson process rainfall model with parameter lambda. We incorporated this context in the introduction and conclusions of the manuscript. We also extended our study to shortwave radiation.

14. Page 6 line 19: Was AT-Neu chosen because it is the first alphabetically? This site is in a north-facing mountain valley and there will be shielding of the sun by mountains to the east and west in the early morning and late afternoon.

Author response: AT-Neu was selected because it has long and high-quality records. You are right about the local orography at this site. That local trait, as well as the Holdridge life zone or Köppen climate zone, must be reflected in the PDF of c and k.

15. Figures S1 to S28 is a bit too much information, even for a supplement. For Fig. 2C of course there is a 180 day negative autocorrelation because of solar geometry. It is interesting to see that c and k have somewhat more complicated long-term autocorrelation functions but is there a better way to syntheize this than to create 28 figures in a Supplement?

Author response: We removed the analysis of ACFs.

16. P. 6 L. 22: Too many of these statements are obvious and follow directly from the solar zenith angle and the amount of atmosphere that a beam has to travel through when the sun is not directly overhead. Also, what does this statement mean 'In these sites, climatic seasonality is very weak since the low ACF after removing the astronomical seasonality.' That the statistics of PAR, c, and k are controlled by solar geometry rather than climate? Of course this isn't surprising but it would be interesting to see that proportion of the variables are explained by climate vs. solar geometry, like a variance decomposition. How much of the variability at each site is explained by these two factors and does Koeppen climate classification help explain some of the variability that is not explained by latitude alone?

    Author response: We removed the analysis of ACFs, made the new Table 2 and the new Figures 3 (a, b and c),and improved the design of Figure 4 (Holdridge triangle) to make more clear the analysis of regionalization of c and k.

17. It is still not clear to me what 'bimodal' means. This is a scale-dependent term. More than one peak per day? More than one peak per season?

    Author response: There was an ambiguity with this term. In climatology it means the occurrence of two dry and two wet seasons along the year (common in the tropics), while in statistics it refers to a PDF with two modes. We removed the ambiguity from the text.

18. The statement on page 7 line 9 isn't supported directly by a figure and I am still confused as to what the major objective of the manuscript is.

    Author response: Results section was rewrote in a more clear form, including disambiguation of the 'bimodality' concept (see answer to comment 17).

19. P 8: reword: allowing to analyze schematically Does Fig. 4 directed at the notion that some sites have darker clouds than others because of the distributions of c and k on wet and dry days?

    Author response: Results section was rewrote in a more clear form, including disambiguation of the 'bimodality' concept (see answer to comment 17).

20. Figure 5: was a Bonferroni correction applied to significance values? Also, please do not simultaneously use red and green in the same figure. Also, why are both KS and AD tests used? What advantages do they each have and why not choose just one? Are the values in the boxes p-values and why are they frequently greater than 1?

    Author response:
    a) No, we didn't because we are not doing several tests at a time. We test all combinations of pairs of months.
    b) We can change the green/red in Fig. 5.
    c) We used the two tests because AD is more sensitive to the tails of the distribution while KS to the center of the distribution. We clarified this point in the text.
    d) The p-values in the boxes are greater than 1 because they are multiplied by 100 to show more decimals using less space. It will be clarified in the legend of the figure.

21. The paragraph after Figure 5 is confusing (p. 11 line 1). I'm not sure what it means: are the data being used to define when seasons begin and end?

Author response: We are comparing the monthly PDFs of c for the dry-days and wet-days samples to inspect for the existence of c and k seasons (i.e. groups of months where c and k have the same PDF). We wrote it more clearly.

22. Figure 6: Please avoid rainbow color schemes (https://eos.org/features/the-end-of-therainbow-color-schemes-for-improved-data-graphics). Also, the relationship between k and c is merely PAR_0 / PAR_cda. This figure only shows how much atmosphere there is which of course is greater at high latitudes in winter when the sun is arriving at an angle (no idea what is happening with US-SRM). It is an inefficient way of showing the effects of the solar zenith angle on surface radiation.

Author response: We removed this figure.

23. I cannot emphasize enough how important it is to have very clear subsections when writing a combined Results and Discussion section. The section jumps surprisingly to different topics throughout and is very difficult to follow. Please add subsections at a minimum to help the reader interpret the flow of the argument. I want to very strongly recommend that the analysis have separate Results and Discussions sections to make it easier to follow and to make the importance of the analysis more clear

Author response: We revised the structure and resolved a problem with Latex. The document is now more clearly structured.

24. Bottom of page 13: I am still not sure what bimodal means in this context and why the analysis is extended to Holdridge life zones. Do some of these ecosystems have expected diurnal or seasonal variability in cloudiness such that grouping the analysis by life zone makes sense?

Author response: We removed the ambiguity from the text (see answer to comment 17).

25. Also, one would expect that a manuscript submitted to Biogeosciences would discuss the importance of the findings to biogeoscience. In this case the role of PAR in controlling photosynthesis is a logical connection. The paper would be stronger if implications for biogeoscience were discussed in more detail.

Author response: We included explicit biogeoscientific context in the introduction and conclusions sections.

**Referee 3**

1. This was an interesting paper about atmospheric attenuation of photosynthetically active radiation (PAR). The paper addresses the spatiotemporal variability in atmospheric attenuation of PAR by analyzing and characterizing the clearness index and the clearday index calculated from long-term observational PAR data for near-globally dispersed sites. The paper provides us with the patterns in atmospheric attenuation of PAR that can be expected for various ecosystems according to their position on the Holdridge triangle or their Köppen climate classification. I enjoyed reading about the indices and the spatiotemporal patterns the researchers have found at a large scale, but the reasons for undertaking the research could be expanded upon.

   Author response: Thank you for your comment. What led us to undertake this research was the need for a probability model of daily radiation to investigate the stochastic dynamics of soil water, nitrogen and carbon contents in energy-limited ecosystems, just as it has been done for water-limited ecosystems (e.g. Ridolfi et al. (2003), Manzoni et al (2004), Botter et al. (2018), Runyan and D'Odorico (2019), Manzoni et al. (2020)). Our ultimate goal is to extend the ecohydrological model of Rodríguez-Iturbe and coauthors from water-limited to energy-limited ecosystems. We are currently working on it (see Muñoz et al. (2020)). We will expand the introduction with this context. We added a final paragraph in the introduction explaining the motivations and importance of our study.

   Botter, G., Daly, E., Porporato, A., Rodríguez-Iturbe, I., & Rinaldo, A. (2008). Probabilistic dynamics of soil nitrate: Coupling of ecohydrological and biogeochemical processes. Water Resources Research, 44(3), n/a-n/a. https://doi.org/10.1029/2007WR006108
   Manzoni, S., Porporato, A., D'Odorico, P., Laio, F., & Rodriguez-Iturbe, I. (2004). Soil nutrient cycles as a nonlinear dynamical system. Nonlinear Processes in Geophysics, 11(5/6), 589–598. https://doi.org/10.5194/npg-11-589-2004
   Manzoni, S., Chakrawal, A., Fischer, T., Schimel, J. P., Porporato, A., & Vico, G. (2020). Rainfall intensification increases the contribution of rewetting pulses to soil respiration. Biogeosciences Discussions, 1--25. https://doi.org/10.5194/bg-2020-95
   Muñoz, E., Ochoa, A., Poveda, G., & Rodríguez-Iturbe, I. (2020). Probabilistic soil moisture dynamics of water- and energy-limited ecosystems. EarthArXiv. https://doi.org/10.31223/osf.io/au4tb
   Ridolfi, L., D'Odorico, P., Porporato, A., & Rodriguez-Iturbe, I. (2003). The influence of stochastic soil moisture dynamics on gaseous emissions of NO, N2O, and N2. Hydrological Sciences Journal, 48(5), 781–798. https://doi.org/10.1623/hysj.48.5.781.51451
   Runyan, C. W., & D'Odorico, P. (2019). Modeling of Phosphorus Dynamics in Dryland Ecosystems. In Dryland Ecohydrology (pp. 309–333). Springer International Publishing. https://doi.org/10.1007/978-3-030-23269-6_12

2. Title: The impression I got from the paper is that it characterizes the site level patterns in atmospheric attenuation that impact how much PAR reaches the ground. The title could be a bit more detailed to include the indices or atmospheric attenuation rather than just "daily PAR".

   Author response: We changed the title to a more precise one.

3. Abstract: The abstract does not communicate why this research was undertaken. The importance of PAR is briefly described in the introduction, but there is no mention of it in the abstract. A sentence about why we should analyze the variability in atmospheric attenuation of PAR in the beginning and another sentence about why the findings or methods are important in the end could help form a complete abstract.

   Author response: We rewrote the abstract in a more clear form.

4. Introduction: At lines 21 and 22, the authors introduce the indices and mention their wide use by other researchers to "quantify the random nature of atmospheric light attenuation" without references to research. The introduction could be expanded to clarify the purpose of studying the variability in atmospheric attenuation of PAR. Some questions below might help expand the introduction: 1. Which studies used the indices to study the variability of atmospheric attenuation? 2. What did those studies find and how does this current research build on previous studies of atmospheric attenuation? 3. Has the variability in the indices been characterized according to climate in the past? If not, why do the authors believe it is important to characterize the variability in atmospheric attenuation by life zone or climate?

Author response: Thank you for your suggestions. We improved the introduction with references and explaining the random nature of c and k and why we expect to find climatic patterns in their statistical properties.

5. Line 12 on pg 6 mentions that the data was separated into rainy and dry days using precipitation. No precipitation dataset is described in the data section. Adding a description of the source for the precipitation dataset will be helpful.

Author response: we got rainfall data also from FLUXNET. This was specified in section 2 (Data).

6. Line 18 on pg 6 says: "The time series, annual cycle, and autocorrelogram of PAR, c and k were calculated and plotted for each site." Is this referring to PAR0 or PARobs?

Author response: Both. We wrote it in a more clear form. The ACFs were removed.

7. It might be helpful to add that the time series, annual cycle, and autocorrelogram were calculated for PAR in the methods section.

Author response: We wrote it in the Methods section (ACF analysis was removed).

8. Figure 2 and the corresponding supplementary figures show what appears to be a confidence interval for the ACF with a dotted line. Which level of confidence does that interval mark?

Author response: It refers to the 95% confidence interval. However, We removed the analysis of ACFs.

9. Figure 2 and figures S1 - S28 need legends with a clarification on which PAR measurement is plotted (PAR0 or PARobs).

Author response: black solid lines in Figure 2(a,b) indicate the PAR for no-atmosphere and thick green solid lines indicate the modeled global radiation. The thin green solid line in Fig. 2(a) is the time series of the observed PAR and the dots in Fig. 2(b) are the mean of PARobs of each day of the year during the record. We will explain this in the legends or captions of Fig. 2 and Figs. S1-S28.

Author's changes in manuscript: The legends of Fig. 2 and Figs. S1-S28 were corrected.

10. It is really hard to read the numbers on the figures with the CDF labeled with numbers (figure 5 and figures S57- S84).

Author response: We changed for symbols instead of numbers.

11. Throughout the paper and figure captions, the parentheses come before the variable they describe. For example: "(a-b) c and (c-d) k". It is a bit easier to read if the variable is mentioned first: "c (a-b) and k (c-d)".

*Author response: it was corrected.*

12. At points in the results/discussion, the figures are introduced by describing the figure. For example: " Fig. 4 shows the PDFs (left panel) and the CDFs (right panel) for wet (blue) and dry (red) days of c (a–b) and k (c–d)." (Pg. 8, line 17). This seems redundant. A good descriptive caption for the figure or a complete legend should take care of this and the text in the results/discussion does not need to mention it.

*Author response: It was corrected.*

13. Line 18 on pg. 8 should read: "Figs. S26 to S56 show the results of the 28 sites analyzed."

*Author response: This was corrected.*

14. Regarding lines 11 - 14 on pg. 7: "We classified the pdfs of c and k in three types: Bimodal, Unimodal I (unimodal with low dispersion), and Unimodal II (unimodal with high dispersion). Sites in the extratropical northern hemisphere (except the site in the United States US-Fep) have bimodal distributions; sites in tropics, subtropics, and USFpe have Unimodal II distributions; and sites in tropics have Unimodal II distributions." This appears to be in disagreement with figure 3. US-Fpe looks like it has a Unimodal I distribution in figure 3.

*Author response: It was an errata. We corrected it and improved the nomenclature for PDF types and the design of the figure with the Holdridge triangle.*

15. If possible, harmonizing the terminology that describes the PDFs between the abstract, results, figures, and conclusion would be helpful. For example, eliminating unimodal I and II altogether and keeping unimodal low and unimodal high to describe the unimodal PDFs throughout the paper and figures should provide consistency for the reader. I also find unimodal low and unimodal high to be more descriptive.

*Author response: We improved the nomenclature for PDF types and revised its use in the whole document. The nomenclature is now: unimodal with low dispersion (ULD), unimodal with high dispersion (UHD), and bimodal (B).*

16. When talking about the PDFs on pg. 7 and 8: The current organization of paragraphs: Discusses the PDFs' latitudinal variability on pg. 7 - top of pg. 8, then talks about the Köppen classification, and then talks about the Holdridge triangle with mention of latitudinal variability. Consider moving the paragraph about the Köppen classification (lines 3 - 7, pg. 8) before mentioning the Holdridge triangle and latitudinal variability so that the discussion on the latitudinal variability is continuous. An order such as: Introduce the classification of the PDFs, then discuss Köppen classification of site PDFs, and then discuss Holdridge triangle position and latitudinal variability of site PDF.

*Author response: Thank you for your suggestion. We reorganized in a more clear form.*

17.    What does "NEP-WCMC" stand for on pg 3 line 12?

Author response: We corrected the typing error ("UNEP-WCMC") and wrote the formal citation: Leemans (1992).

18.    There seems to be some disagreement between the abstract and the conclusion. The abstract says: "Unimodal distributions with high dispersion are concentrated in the moist forest life zone in subtropical and tropical regions and humid province; and unimodal distributions with low dispersion are concentrated in dry forest, very dry forest, and thorn woodland in tropical and subtropical regions between arid and subhumid humidity provinces." The conclusion says: "High latitudes sites exhibit bimodal distributions, arid to sub- humid climates exhibit unimodal distributions with high dispersion, and humid tropical regions exhibit unimodal distributions with low dispersion."

Author response: Author response: We revised the coherence in the whole manuscript.

[revised manuscript text omitted]

$$PARH0_0 = \frac{24E_0}{\pi} TSI_{PARBand} (\omega_{sr} \sin\delta \sin\phi + \cos\delta \cos\phi \sin\omega_{sr}) \tag{3}$$

where $E_0$ is the eccentricity correction factor of the earth's orbit and $\omega_{sr}$ is the sunrise hour angle for the day.

**3.2  Daily surface radiation for a cloudless, clean dry atmosphere**

 Daily surface radiation for a cloudless, clean, and dry atmosphere ($ H_{cda}$) is the sum of the direct ($ H_b$) and diffuse ($  $PAR_{cda} = PAR_b + PAR_d)H_d$) components. To calculate daily $$ and $ H_b$ and $H_d$ on a horizontal surface at the ground level, we model the direct and diffuse instantaneous spectral irradi-

5 ances and integrate them along the day length and the PAR  and SW spectral domains.

Following Iqbal (1983), we assume the cloudless, clean and, dry atmosphere to be composed by uniformly mixed gases (m) and ozone (o). Using the Beer–Lambert law and integrating, daily $ H_b$ is calculated as in Eq. (4).

$$ R_b = \int\limits_{\gamma_{sr}}^{\gamma_{ss}} \int \frac{700\ nm}{400\ nm}_{Band} SSI_{0,n,\lambda} E_0 \sin(\gamma)\tau_{ma,\lambda}\,\mathrm{d}\lambda\,\mathrm{d}\gamma \qquad (4)$$

where $SSI_{0,n,\lambda}$ is the extraterrestrial spectral irradiance normal to the rays from the sun (obtained from SOLID), $\gamma$ is the

10 solar altitude varying from sunrise ($sr$) to sunset ($ss$), and $\tau_{ma,\lambda}$ is the transmittance due to the molecular absorbers of the cda atmosphere.

For the assumed atmosphere composition $\tau_{ma,\lambda} = \tau_o \cdot \tau_g$, where $\tau_o$ and $\tau_g$ are the ozone and the mixed gases transmittance, respectively (see details in Iqbal, 1983, Sec.6.14).  We arbitrarily assumed forward and backward scatterances of $0.5$ and  considered only the first pass of radiation through the atmosphere, . Although higher reflectances

15 could bring about some subestimation of $H$, especially during snow-cover periods, we think it is not a critical issue for the sake of this study. Uncertainty caused by these two assumptions will be included in the statistical properties of $c$. $H_d$ can then be calculated by

$$ H_d = \int\limits_{\gamma_{sr}}^{\gamma_{ss}} \int \frac{700\ nm}{400\ nm}_{Band} SSI_{0,n,\lambda} E_0 \sin(\gamma)\tau_{ma,\lambda}[0.5(1-\tau_{r,\lambda})]\,\mathrm{d}\lambda\,\mathrm{d}\gamma \qquad (5)$$

where $\tau_{r,\lambda}$ is the transmittance due to Rayleigh molecular scattering (see details in Iqbal, 1983, Sec.6.14).

20 Several atmospheric parameters are required by Eqs. (4) and (5). We assume the 1976 U.S. standard atmosphere (NASA, 1976) (sea level pressure of 101.325 kPa, sea level temperature of 288 K, and sea level density of 1.225 kg/m$^3$) and the Kasten and Young (1989, Table II) optical air mass function of solar altitude, which has 336 values for solar altitudes between $0°$ and $90°$. Transmittance for ozone and mixed gases are calculated as in Eqs. (6) to (8).

$$\tau_{o,\lambda} = \exp(-k_{o,\lambda}l_o m_r) \qquad (6)$$

$$\tau_{g,\lambda} = \exp\left[\frac{-1.41k_{g\lambda}m_a}{(1+118.93k_{g\lambda}m_a)^{0.45}}\right] \qquad (7)$$

[revised manuscript text omitted]

---

## Author Response (AR2)

Dear Paul C. Stoy
Editor Biogeosciences

We acknowledge the opportunity to submit another revised draft of our manuscript. We have implemented the suggestions made by the referee. Please see below a point-by-point response and the changes made as a consequence of these suggestions.

Sincerely,
Estefanía Muñoz and Andrés Ochoa

1. Abstract: it would still help (me at least) to define in the abstract if hourly (or half-hourly) or daily PAR and SW values are used in the analysis to help communicate what is being displayed in the PDF.

   Author response: Line 4 in the Abstract says that we use daily data.

2. Rodriguez-Iturbe and colleagues do great research; this is true. But the text as written focuses on their research instead of yours and how your work pushes forward stochastic analysis in Earth science. Also, as a consequence, the importance of your work is subsumed by the work of others. It's interesting for a number of applications including for example solar power forecasting to understand the stochastic nature of atmospheric attenuation of radiation. The purpose of the present paper is to explain how this (partly) random process differs across the globe. Focusing on this novelty would improve the Introduction.

   Author response: We modified the introduction shortening the description of Rodríguz-Iturbe and collaborators' work and generalizing the application of our research.

3. Note a few minor typos like the lines above equation 2.

   Author response: We corrected the typos throughout the manuscript.

4. p. 6: define doy as day of year.

   Author response: It was done in line 14 of page 6.

5. I recommend a proofread before final submission to catch things like 'more sensitive to the tails KS' (should be 'tails of KS'), and 'WS' in the conclusions, etc.

Author response: We did a proofread of the whole document.

6. Figure 5 and 6 is convincing, but a formal categorization analysis of the different pdf types would be a bit more robust. The data do seem to sort nicely with some interesting outliers.

Author response: We agree that an objective and automatic method of categorization would be more robust. At this point of the process, however, it implies a large amount of reprocessing that we don't think it's worth it. Although our visual method is off course subjective, we did it carefully and are confident about the results.

7. Instead of 'neater than' it may be better-written 'arguably more effective at classifying sites based on their solar radiation characteristics'.

Author response: It was changed in the first paragraph of the discussion.

8. DE-Geb, and DE-Hai are very close to each other, I am curious to know the mechanism by which dry days still have bimodal distributions? Is fog an issue?

Author response: We included a short analysis of the local climate in this sites in the second paragraph of Discussion.

[revised manuscript text omitted]